# Characterization of Two Porcine Parainfluenza Virus 1 Isolates and Human Parainfluenza Virus 1 Infection in Weaned Nursery Pigs

**DOI:** 10.3390/vetsci10010018

**Published:** 2022-12-28

**Authors:** Michael Welch, Karen Krueger, Jianqiang Zhang, Megan Neveau, Pablo Piñeyro, Drew Magstadt, Rodger Main, Phillip Gauger

**Affiliations:** Department of Veterinary Diagnostic and Production Animal Medicine, Iowa State University, Ames, IA 50011, USA

**Keywords:** porcine parainfluenza virus 1, human parainfluenza virus 1, *Paramyxoviridae*, *Respirovirus*, strain comparison, pathogenesis

## Abstract

**Simple Summary:**

Porcine respiratory disease is responsible for high economic costs to swine producers. The porcine respiratory disease complex is composed of multiple primary and secondary viral and bacterial pathogens, respectively. Porcine parainfluenza virus 1 (PPIV1) is an emerging respiratory virus that has been detected in all ages of swine. However, it is unknown if PPIV1 is a component of the porcine respiratory disease complex or if different strains of the virus may be responsible for inducing clinical respiratory disease in pigs. A study was conducted using two different strains of PPIV1 and a human parainfluenza virus 1 (HPIV1) to evaluate differences in clinical disease and virus replication, shedding and lung lesions. A group of 24 conventional nursery pigs were challenged with PPIV1 MN16, PPIV1 IA17 or HPIV1 and a negative control group of 6 pigs was included. Overall, average daily gain was similar among the groups when evaluated at 28 days post inoculation (DPI). Virus replication and shedding was also similar in the MN16 and IA17 groups although much lower in the HPIV1 group. However, the IA17 pigs demonstrated higher amounts of lung damage compared to the other treatment groups. All challenged pigs regardless of the group seroconverted. These findings indicate that PPIV1 clinical respiratory disease may notbe affected by different strains of PPIV1 although pigs can be infected with a human parainfluenza virus. This study suggests more experiments are needed to evaluate additional strains of PPIV1 and their potential to cause respiratory disease in swine.

**Abstract:**

Porcine parainfluenza virus 1 (PPIV1) is a newly characterized porcine respiratory virus. Recent experimental challenge studies in three-week-old nursery pigs failed to cause disease. However, it remains unclear how genetic differences contribute to viral pathogenesis. To characterize the pathogenesis of different PPIV1 isolates, three-week-old nursery pigs were challenged with either PPIV1 isolate USA/MN25890NS/2016 (MN16) or USA/IA84915LG/2017 (IA17). A human parainfluenza virus 1 (HPIV1) strain C35 ATCC^®^ VR-94™ was included to evaluate swine as a model for human parainfluenza. All viruses were successfully re-isolated from bronchoalveolar lavage fluid and detected by RT-qPCR at necropsy. Microscopic lung lesions were more severe in the IA17 group compared to the non-challenged negative control (Ctrl) group whereas differences were not found between the MN16 and Ctrl groups. Immunohistochemistry staining in respiratory samples showed a consistent trend of higher levels of PPIV1 signal in the IA17 group followed by the MN16 group, and no PPIV1 signal observed in the HPIV1 or Ctrl groups. This study suggests potential pathogenesis differences between PPIV1 isolates. Additionally, these results indicate that HPIV1 is capable of replicating in nursery pigs after experimental inoculation. However, clinical disease or gross lung lesions were not observed in any of the challenge groups.

## 1. Introduction

Porcine parainfluenza virus 1 (PPIV1) is a newly characterized swine respiratory virus in the family *Paramyxoviridae*, subfamily *Orthoparamyxovirinae*, and genus *Respirovirus.* Porcine parainfluenza virus 1 is a single stranded, negative sense RNA virus approximately 15–15.5 kilobases in length. There are six open reading frames in all Paramyxoviruses that sequentially encode from 3′ to 5′ for a nucleocapsid, phosphoprotein polygene, matrix protein, fusion protein, hemagglutinin-neuraminidase protein and large polymerase protein. The hemagglutinin-neuraminidase and fusion genes are commonly used for tracking viral epidemiology as they contain many important neutralizing epitopes [1,2]. Other related viruses include Sendai virus (SeV), human parainfluenza virus 1 (HPIV1), bovine parainfluenza virus 3 (BPIV3), caprine parainfluenza virus 3 (CPIV3), and giant squirrel parainfluenza virus (SPIV) [3]. Both HPIV1 and BPIV3 are established respiratory pathogens in humans and cattle, respectively. The human parainfluenza viruses cause relatively few clinical signs in healthy adults but are associated with more severe clinical disease in infants and immunocompromised patients [4,5,6,7].

PPIV1 has been detected worldwide in Asia [8], South America [9], North America [10,11], and Europe [12,13]. Published whole-genome sequences suggest some degree of genetic diversity between PPIV1 isolates globally. Whole-genome sequences from North American strains share from 95.7% to 98.1% nucleotide homology and form separate monophyletic clades from PPIV1 sequences detected in Hong Kong in 2013 [8,11]. In contrast, whole-genome or F-gene sequences from South America and Europe cluster more closely with those from Hong Kong [9,12,13]. Phylogenetic analysis of North American PPIV1 remains limited and additional sequences are necessary to evaluate the breadth of PPIV1 genetic diversity in strains circulating in swine in the United States (U.S.).

Challenge studies using 3–4 week old conventional pigs or 5–6 week old cesarean derived colostrum deprived (CDCD) pigs have described PPIV1 replication in bronchi and bronchiolar, trachea, and nasal turbinate (NT) epithelium in the absence of clinical disease or gross lesions [14]. Microscopic lung lesions are characterized by epithelial attenuation and mild to moderate peribronchiolar cuffing consisting of mononuclear cells [14]. However, it remains unknown if genetically different strains of PPIV1 may have a different phenotype and may be responsible for potential differences in the clinical presentation and severity of lesions observed between experimental and naturally infected pigs [15]. This study investigates the pathogenicity, replication kinetics, and antibody response of two PPIV1 cell culture isolates and one HPIV1 isolate to evaluate phenotype differences between strains.

## 2. Materials and Methods

### 2.1. Viruses and Cell Culture

All parainfluenza viruses used in this study, including the PPIV1 isolate USA/MN25890NS/2016 (MN16, passage 9), isolated in 2016 from nasal swabs (NS) collected from suckling pigs with respiratory disease, USA/IA84915LG/2017 (IA17, passage 5), isolated in 2017 from the lung of nursery pigs with coughing, and the HPIV1 isolate (Strain C35 ATCC^®^ VR-94™) were passaged in *Macaca mulatta* kidney cells (LLC-MK2, ATCC^®^ CCL-7™) as previously described [11]. Briefly, cells were maintained in M199 medium (Gibco^®^, Waltham, MA, USA) supplemented with 1% equine serum (Sigma Aldrich, St. Louis, MO, USA) and 1% penicillin-streptomycin antibiotics (Thermo Fisher Scientific, Waltham, MA, USA). At approximately 95% confluency, cells were dissociated with 0.05% ethylenediaminetetraacetic (EDTA) trypsin (Gibco, Waltham, MA, USA) for one minute and centrifuged at 125× *g* for five minutes before seeded to flasks or 96 well Microtiter™ plates (Thermo Fisher Scientific, Waltham, MA, USA) at 37 °C at 5% CO_2_.

The MN16, IA17, and HPIV viruses were inoculated in T75 flasks (Thermo Fisher Scientific, Waltham, MA, USA) at a multiplicity of infection of 0.01 in post inoculation medium consisting of Earle’s minimum essential medium (MEM, Gibco, Waltham, MA, USA) supplemented with 1 μg/mL tosyl phenylalanyl chloromethyl ketone (TPCK) treated trypsin, 1% penicillin, 1% streptomycin, 0.1% amphotericin B, and 0.1% gentamicin (Thermo Fisher Scientific, Waltham, MA, USA). The virus was adsorbed for 2 h at 37 °C before being decanted and replaced with fresh PIM. The virus was incubated for 48–72 h before the flasks were frozen and thawed. The cell debris were clarified by centrifugation at 4200× *g* for 10 min.

Virus titrations were performed in 96 well Microtiter™ plates by serial dilution. Virus isolates were diluted 1:10 in post inoculation medium in quintuplicate, mixed, starting at 10^1^ to 10^8^. Cells were washed 3 times with 125 μL post inoculation medium. Exactly 100 µL of the virus dilution was transferred to the confluent cells in 96 well plates and allowed to incubate for 72 h. The plates were fixed with 100 µL of 80% acetone at −20 °C and stained using 1:50 dilution of PPIV1 primary monoclonal antibody mixture targeting PPIV1 proteins and developed in mice as previously described by Jie Park et al. [11], 1:200 dilution of biotinylated goat anti-mouse IgG secondary antibody (Thermo Fisher Scientific, Waltham, MA, USA), and 1:100 dilution of streptavidin conjugated horseradish peroxidase (HRP) (Thermo Fisher Scientific, Waltham, MA, USA) [11]. All antibody steps were incubated for 1 h at 37 °C. Antibody dilutions were performed in phosphate-buffered saline (PBS, Gibco, Waltham, MA, USA) with 0.5% tween-20 (PBST) containing 1% bovine serum albumen (BSA). The cells were washed 4 times with 200 µL of PBST between each antibody application. Next, 100 µL of aminoethyl carbazole (AEC) chromogen was incubated on the cells until sufficient color change was visualized (<10 min) and a final wash with deionized water.

PPIV1 and HPIV1 virus re-isolation from experimental samples were performed on bronchoalveolar lavage fluid (BALF) as previously described on LLC-MK2 cells with slight modifications [11]. Briefly, BALF samples were diluted two-fold in post inoculation medium, the cells were washed 3 times with post inoculation medium, and 100 µL was added in quintuplicate to 96 well plates. The virus was allowed to adsorb on the cells for 2 h and the inoculum was replaced with an equal amount (100 µL) of post inoculation medium. The cells were incubated for 72 h, fixed with 80% acetone at −20 °C, and PPIV1 stained as described above. The HPIV immunocytochemistry stain was conducted using a primary mouse HPIV1 antibody (Santa Cruz Biotechnology Inc., Dallas, TX, USA), diluted 1:100 in PBST with 1% BSA, and incubated for 1 h. The plates were washed 4 times in PBST and 1:200 dilution of goat anti-mouse polyclonal antibody conjugated to HRP was incubated for 1 h. The plates were washed 4 times in PBST and the reaction was visualized with AEC chromogen as described above.

### 2.2. Whole-Genome Sequencing

Whole-genome sequencing of USA/MN25890NS/2016 and USA/IA84915LG/2017 PPIV1 isolates was conducted at the Iowa State University Veterinary Diagnostic Laboratory (ISU VDL) next generation sequencing department using the Illumina MiSeq as previously described [11,16,17]. The complete genomes have been deposited in GenBank under the accession number MF681710 for MN16 and MG753974 for IA17 (Table 1).

### 2.3. Phylogenetic Analysis

Alignment of 113 parainfluenza virus (PIV) whole-genome sequences available in GenBank was performed in Geneious (v2021.2) with the MAFFT algorithm (v7.450) using default settings. The sequences were comprised of 76 HPIV1, 10 PPIV1, 1 SPIV, 26 SeV, 1 BPIV3, and 1 CPIV3. A maximum likelihood (ML) phylogenetic tree was constructed with RAxML (v8) and visualized in FigTree (v1.4.4). The final ML dendrogram was subsampled by 90% while preserving overall genetic diversity (https://github.com/flu-crew/smot; accessed on 10 September 2022).

### 2.4. PPIV1 Antibody Detection by wv-ELISA

PPIV1 antibodies were detected by a whole virus enzyme linked immunosorbent assay (wv-ELISA) conducted at the ISU VDL as previously described [18]. Briefly, 500 mL of viable MN16 isolate solution was prepared in cell culture as described above to a titer of approximately1 × 10^6^ to 1 × 10^7^ 50% tissue culture infectious dose (TCID_50_) per ml (TCID_50_/mL). The clarified MN16 PPIV1 supernatant was ultracentrifuged at 140,992× *g* for 3 h, washed twice with 50 mL of PBS pH 7.4 and the final pellet of concentrated virus was resuspended with 100 µL of PBS. The optimum dilution of concentrated PPIV1 was determined using a checkerboard titration based on known antibody positive and negative sera to maximize signal while minimizing background noise. Polystyrene 96 well plates (Nunc, Maxisorp, Thermo Fisher Scientific, Agawam, MA, USA) were coated with 100 µL of the whole virus solution at an optimum dilution (1:200 in PBS) per well, washed five times with PBST (PBS w/0.1% Tween 20), and blocked with 1% (*w*/*v*) BSA (Jackson ImmunoResearch, West Grove, PA, USA). The plates were dried and stored for later use with desiccant packs.

Serum samples were diluted at 1:100 in diluent containing 40% newborn calf serum (Gibco^®^, Waltham, MA, USA), added to the antigen-coated plates, and incubated at room temperature for 1 h. Plates were washed five times with PBST prior to adding 100 µL of goat anti-porcine secondary antibody conjugated to HRP (Bethyl Laboratories Inc., Montgomery, TX, USA) at 1:20,000 dilution and incubated for 1 h at room temperature. After 5 x wash with PBST, the reaction was visualized with tetramethylbenzidine-hydrogen peroxide (TMB, Surmodics IVD Inc., Eden Prairie, MN, USA). The reaction was stopped by adding 100 μL stop solution (Surmodics IVD Inc., Eden Prairie, MN, USA) to each well and read at an optical density (OD) of 450 nm. A PPIV1 antibody positive and negative serum sample collected from pigs of known status from prior PPIV1 challenge studies were included on each plate and tested in duplicate to establish positive control and negative control mean OD values for calculating the sample to positive (S/P) ratio [14]. Anti-PPIV1 wv-ELISA antibody S/P ratios were calculated per the equation:$$S/P\;\mathrm{Ratio}:\frac{\left(\mathrm{sample}\;\mathrm{OD}-\mathrm{negative}\;\mathrm{control}\;\mathrm{mean}\;\mathrm{OD}\right)}{\left(\mathrm{positive}\;\mathrm{control}\;\mathrm{mean}\;\mathrm{OD}-\mathrm{negative}\;\mathrm{control}\;\mathrm{mean}\;\mathrm{OD}\right)}$$

### 2.5. HPIV1 Antibody Detection by ELISA

Antibody levels against HPIV1 were measured using a human HPIV1-3 commercial assay (GenWay Biotech, San Diego, CA, USA) adapted for porcine serum by replacing the anti-human conjugated HRP (GenWay Biotech, San Diego, CA, USA) secondary antibody with anti-porcine HRP antibody. After optimization using a subset of positive and negative samples, dilutions of 1:25 of porcine samples and 1:10,000 of secondary antibody (Bethyl Laboratories Inc., Montgomery, TX, USA) were chosen for use in the assay. All other parameters in the assay were followed as outlined in the instruction manual.

### 2.6. PPIV1 Serum Virus Neutralization (SVN)

Neutralizing antibodies against PPIV1 were quantified as previously described [14]. Briefly, sera were diluted in post inoculation medium starting at 1:10 to 1:1280. The MN16 isolate was diluted to a concentration of 4000 TCID_50_/mL (200 TCID_50_/50 μL) before addition to an equal volume of serially diluted sera. The mixture was incubated for 60 min at 37 °C and 100 µL transferred to confluent monolayers of LLC-MK2 in 96 well plates for 2 h at 37 °C and discarded before washing the cells 2 times with MEM and adding 100 µL post inoculation medium. The plates were incubated for 72 h, fixed with 80% acetone at −20 °C, and immunocytochemistry detection was performed as described above.

### 2.7. Animal Study Design

Sows were screened on the farm for PPIV1 antibody using the wv-ELISA at the ISU VDL and pigs were subsequently selected from antibody negative sows for additional screening. Nasal swabs were collected from pigs at the farm and tested for PPIV1 RNA by RT-rtPCR and sera were tested by wv-ELISA for presence of PPIV1 per ISU VDL standard operating procedure. After arrival to the research facility at Iowa State University, pigs were additionally screened by wv-ELISA for PPIV1 antibody and PCR for PPIV1, influenza A virus in swine, and *Mycoplasma hyopneumoniae* in NS. Sera were screened for porcine reproductive and respiratory syndrome virus and porcine circovirus type 2 viremia to confirm negative status. Pigs were randomized by weight, assigned to treatment groups at −3 days post inoculation (DPI) and acclimated for three days prior to challenge. Six pigs were assigned to the non-challenged negative control (Ctrl) group and eight pigs each were assigned to the MN16, IA17, and HPIV1 challenge groups (Table 1). The challenge pigs were inoculated with 2 mL intratracheally and 2 mL intranasally divided evenly between each nostril at an estimated titer of 1 × 10^5^ TCID_50_/mL of their respective challenge viruses.

Five pigs from each challenge group and four pigs from the Ctrl group were randomly selected and euthanized for necropsy at 5 DPI. Three pigs from each challenge group and two from the Ctrl group were necropsied at 28 DPI (Table 1). Weights were collected at randomization step and at each necropsy and average daily gain was calculated based on time from initial weight collection. Fresh samples collected at necropsy included tracheal mucosal swabs (TS) in 2 mL MEM (Thermo Fisher Scientific, Waltham, MA, USA), cranioventral lobes of lung, BALF, and NT. Fixed tissues collected in 10% formalin included lung, trachea and NT. The NS were collected in 2 ml PBS and body temperatures were collected daily from 0–7 DPI, and additional NS were collected weekly at 14, 21, and 28 DPI. Sera were collected at 0, 7, 10, 14, 17, 21, and 28 DPI. Coughing (absent/present) and respiratory scores (0: normal, 1: mild but increased respiration when active, 2: moderate increased respiration at rest with mild dyspnea when active, 3: severe, constant, dyspnea, and abdominal breathing or open-mouth breathing) were also recorded at 0–7, 10 and 14 DPI.

### 2.8. Sample Processing

All samples were processed as previously described and per ISU VDL protocols. The NT were processed with a disposable tissue grinder system (Thermo Fisher Scientific, Waltham, MA, USA). A 5% (*w*/*v*) homogenate was made by adding 0.5 g of tissue to 10 mL of Earle’s salts and grinding for 30 s. A 10% (*w*/*v*) lung homogenate solution was created by placing 1 g of tissue into 10 mL Earle’s salts solution and homogenizing the tissue in a Stomacher^®^ 80 Biomaster (Thermo Fisher Scientific, Waltham, MA, USA) for 3 min. The NS and TS did not require further processing prior to nucleic acid extraction.

### 2.9. Nucleic Acid Extraction and Detection of PPIV1 and HPIV1 by RT-qPCR

Viral RNA was extracted from processed lung, BALF, TS, and NT using the 5 X Ambion^®^ MagMax™ 96 viral RNA kit (Thermo Fisher Scientific, Waltham, MA, USA) and automated Kingfisher 96^®^ magnetic particle processor (Thermo Fisher Scientific, Waltham, MA, USA) as specified by manufacturer protocols.

PPIV1 RNA was detected by quantitative reverse transcription PCR (RT-qPCR) and analyzed as previously described [11] using standard curve quantitation developed at the ISU VDL based on a synthesized and quantified PPIV1 template sequence to provide genomic copies/mL (GC/mL). Primers targeting the nucleocapsid gene of PPIV1 were obtained from Integrated DNA Technologies (IDT Technologies, Coralville, IA, USA). Cycle thresholds (Ct) equal to or above 40 were considered negative. The probe contained a 5′FAM^®^ fluorophore and 3′Iowa Black^®^ quencher. Signal amplification was performed and monitored using a 7500 Fast thermocycler (Applied Biosystems, Foster City, CA, USA). Results were analyzed with commercial software and reported as GC/mL of their respective processed sample or homogenate.

An isolate-specific prototype RT-qPCR was developed to target the F gene of HPIV1 using primers and probes developed in Primer Express™ software v3.0.1 (Thermo Fisher Scientific, Waltham, MA, USA) (Table 2) using identical cycling conditions to the PPIV1 RT-qPCR. Signal amplification and monitoring was conducted using a 7500 Fast thermocycler using an auto-baseline and threshold setting at 10% of the maximum signal amplitude. The probe contained a 5′FAM^®^ fluorophore and 3′Iowa Black^®^ quencher. A Ct value ≥ 36 was considered negative. A standard curve was developed using synthesized template as described above [19] and viral quantities were reported as GC/mL.

### 2.10. Macroscopic and Microscopic Lesions and Immunohistochemistry Scores

Macroscopic lung lesions were evaluated by a veterinary pathologist blinded to treatment groups. The weighted proportions of affected lung lobes based on percent of cranioventral consolidation were added relative to the lung volume as previously described for porcine reproductive and respiratory syndrome virus [20] and previous PPIV1 pathogenesis studies [14]. Microscopic lung and trachea lesions and PPIV1 immunohistochemistry (IHC) signal in lung, trachea and NT were evaluated by a veterinary pathologist blinded to treatment groups and based on previous PPIV1 pathogenesis studies [14]. Composite lung lesion scores ranged from 0–22 and included individual scores of peribronchiolar cuffing (0–4), airway epithelial necrosis (0–4), suppurative bronchiolitis (0–4), epithelial microabscesses (0–3), interstitial pneumonia (0–4), and alveolar edema (0–3). The composite trachea score ranged from 0–8 and consisted of trachea epithelial necrosis (0–4) and tracheitis (0–4). Composite PPIV1 IHC scores in the lung ranged from 0–8 and consisted of airway epithelium signal (0–4) and interstitial signal (0–4) and the trachea and NT scores ranged from 0–4 consisting of tracheal or NT signal in the mucosal epithelium.

### 2.11. Statistical Analysis

Analysis of NS and necropsy sample RT-qPCR results, SVN titers, average daily gain, HPIV ELISA OD values, PPIV wv-ELISA S/P ratios, and body temperatures were analyzed using a linear mixed regression model (PROC MIXED) (SAS version 9.4, SAS Institute Inc., Cary, NC, USA). The most appropriate covariance structure was chosen for each model, respectively, based on the lowest corrected Akaike’s information criterion value. The ELISA S/P and OD values for PPIV and HPIV, respectively, were natural log transformed for normality. The RT-qPCR GC/mL values were log_10_ transformed prior to fitting the model. Due to the significant amount of left censoring observed in the NS RT-qPCR results, a tobit model (PROC LIFREG) was fitted to account for values below the limit of detection. Median microscopic lesion scores and median IHC scores were analyzed with a Kruskal–Wallace test and Dwass, Steel, Chritchlow–Flinger multiple comparison analysis if significant (PROC NPAR1WAY). Macroscopic lung lesions were minimal and statistical comparisons were not performed.

## 3. Results

### 3.1. Phylogenetic Analysis

The HPIV1, SeV, SPIV, and PPIV1 whole-genome sequences formed distinct monophyletic clades (Figure 1). The MN16 and IA17 isolates used in this study shared 98.19% nucleotide identity and clustered closely together. In contrast, the HPIV1 isolate shared 64.33% and 64.21% identity with the MN16 and IA17 isolates, respectively. Within the PPIV1 whole-genome sequences, nucleotide homology ranged from 90.90% to 98.18%. HPIV1 sequences ranged from 93.82% to 99.99% and SeV ranged from 86.45% to 99.99%.

Both the IA17 and MN16 shared a high degree of homology with only 1.49% and 3.07% difference across individual genes. Homology between the PPIV1 and HPIV1 strains were much lower as expected. The highest genetic diversity occurred on the surface glycoproteins (HN, F) as well as the P protein while the nucleoprotein tended to be the most conserved between HPIV1 and PPIV1 isolates used in this study (Table 3).

### 3.2. Clinical Performance and Average Daily Gain

No significant difference was observed in average daily gain at 5 DPI necropsy (−3 to 5 DPI) between the Ctrl, HPIV1, and IA17 groups (Figure 2A). Interestingly, pigs challenged with MN16 showed a significantly decreased average daily gain compared with the Ctrl and IA17 group (*p* < 0.02). However, significant differences in average daily gain were not observed in Ctrl, HPIV1, IA17 and MN16 challenged pigs at 28 DPI (−3 to 28 DPI, Figure 2B). No significant respiratory distress nor coughing was observed in any of the challenge groups.

### 3.3. Viral Shedding Detection in Nasal Swabs

Viral shedding in NS was detected in 100% (8/8) pigs in the IA17 and MN16 group at 1 DPI demonstrating an average of 4.09 and 4.65 log_10_ GC/mL, respectively (Figure 3A). Pigs in the MN16 group shed significantly more virus at 1 DPI compared to the IA17 group (*p* = 0.023). Shedding was detected at high levels from challenged pigs regardless of PPIV1 isolate at 2–7 DPI and was markedly lower at 14 DPI. No significant differences were observed between the MN16 and IA17 GC/mL at 2-28 DPI. Detection of HPIV1 by RT-qPCR was variable and consisted of 25% (2/8) or 88% (7/8) pigs testing positive at various times 1–5 DPI and 34% (1/3) pigs positive at 7 DPI. HPIV1 RNA quantities ranged from 2.78–4.96 log_10_ GC/mL from actively shedding pigs. No viral RNA for HPIV or PPIV were detected in the Ctrl group throughout the study.

### 3.4. Virus Detection in Tissues

Viral RNA was detected from all pigs in the HPIV1, IA17, and MN16 groups at 5 DPI in all sample types (Figure 3B). Consistently higher levels of RNA were detected in the IA17 and MN16 groups compared to the HPIV1 pigs with evidence of an interaction between sample type and treatment (*p* = 0.06). Significant differences in viral GC/mL were not observed between IA17 and MN16 challenged pigs regardless of sample type. A 2.82 log_10_ (95% CI 2.58–3.07) and 2.62 log_10_ (95% CI 2.36–2.87) marginal difference in viral GC/mL were observed between the HPIV group and the IA17 and MN16 groups, respectively (Figure 3B). Detection of HPIV1 was more consistent in tissues compared to NS with HPIV1 RNA detected from all pigs and sample types at greater than 3.21 log_10_ GC/mL (Figure 3B).

Virus was successfully isolated from 60% (3/5) of HPIV1 pigs, 100% (5/5) of IA17 pigs, and 75% (3/4) of MN16 pigs from BALF samples collected at 5 DPI. Animals were considered positive if virus was successfully isolated from at least one well out of four replicates based on expected low concentrations of virus in the sample type. No virus was isolated from the BALF collected from the Ctrl pigs at 5 DPI.

### 3.5. Porcine and Human Parainfluenza Virus Humoral Antibody Response

Neutralizing antibody was observed as early as 7 DPI in both MN16 and IA17 groups compared to the Ctrl group (*p* < 0.001) (Figure 4A). Pigs from the MN16 group demonstrated significantly higher SVN antibody titers compared to the IA17 group on 7 and 10 DPI (*p* < 0.001). In contrast, SVN titers were significantly higher in the IA17 group at 24 DPI compared to the MN16 group (*p* = 0.02). No PPIV1 neutralizing antibody was detected in the Ctrl or HPIV1 groups throughout the study.

At 7 DPI, the mean PPIV1 wv-ELISA S/P ratios in the Ctrl group were not significantly higher than the MN16 group (*p* = 0.066) but paradoxically were significantly greater than the IA17 (*p* = 0.0014) and HPIV1 (*p* = 0.0093) groups (Figure 4B). However, the absolute difference was small between the highest (Ctrl) and lowest (IA17) groups at a difference in S/P ratio of 0.036. The IA17 group also had significantly elevated average S/P ratios on 21 and 24 DPI relative to all other groups (*p* < 0.05). No significant differences in S/P ratio were observed between the IA17 and MN16 groups on 10, 14, 17, and 28 DPI (*p* < 0.05). Antibody from HPIV1 challenged pigs did not appear to cross react with PPIV1 antigen from 0–28 DPI (*p* < 0.05) by the PPIV1 wv-ELISA, and PPIV1 wv-ELISA antibodies were not detected in the Ctrl group throughout the study.

In pigs included in the HPIV1 group, antibodies were detected by the HPIV1-3 ELISA from 14-24 DPI and OD values were significantly higher compared to the Ctrl group (*p* < 0.001) (Figure 4C). Interestingly, significantly higher HPIV OD values (*p* < 0.003) were also observed in the IA17 group relative to the Ctrl group on 14–24 DPI. No significant difference in HPIV1 antibody levels was observed between MN16 and the Ctrl group (Figure 4C). 

### 3.6. Macroscopic and Microscopic Lesions at 5 DPI

Gross lung lesions were not observed in either the Ctrl or any of the challenged groups (data not shown). None or minimal histologic changes in the trachea were observed in the Ctrl and HPIV groups at 5 DPI (Figure 5A). There is weak evidence of a statistical difference in median microscopic trachea lesion scores between the IA17 and Ctrl (*p* = 0.048), MN16 and Ctrl (*p* = 0.061) and IA17 and HPIV1 groups (*p* = 0.058). In contrast, no significant difference was observed between median IA17 and MN16 trachea lesion scores (*p* > 0.1). Microscopic trachea epithelium appeared unaffected in Ctrl (Figure 6A) or minimally affected in the HPIV1 pigs (Figure 6B). A loss of cilia and goblet cells was observed in the MN16 (Figure 6C) and IA17 (Figure 6D) groups including attenuation of the tracheal epithelium. The submucosa was mildly expanded by macrophages and lymphocytes. 

Mild microscopic lung lesions were observed in the HPIV, MN16 and IA17 groups. There was inconclusive evidence of a difference in microscopic lesion scores between the IA17 group compared to the Ctrl group (*p* = 0.060) and that of the MN16 and IA17 groups (*p* = 0.065; Figure 5B). No significant differences in lung lesion scores were observed between the MN16, HPIV, and Ctrl groups (*p* > 0.1, Figure 5B). Microscopic lesions were not observed in the Ctrl group lung (Figure 6E). Mild peribronchiolar cuffing was observed in 1 pig in the HPIV1 group (Figure 6F) in contrast to the lung lesions observed in the MN16 (Figure 6G) and IA17 (Figure 6H) groups that were characterized by epithelial proliferation and mild peribronchiolar cuffing with mononuclear cells.

### 3.7. Immunohistochemistry Results at 5 DPI

A consistent trend was observed in the lung, trachea, and NT where higher levels of PPIV1 signal was observed in the IA17 (Figure 6K) group followed by the MN16 (Figure 6L) group, and no PPIV1 signal was observed in the HPIV1 (NT IHC data not included, Figure 5C,E and Figure 6J) and Ctrl groups (Figure 5C–E and Figure 6I). There is moderate evidence to indicate a significant difference in median IHC scores between the MN16 and IA17 groups in both the trachea (*p* = 0.024) (Figure 5C) and NT (*p* = 0.027) (Figure 5D); however, results comparing IHC scores between the MN16 and IA17 lung were suggestive of a significant but inconclusive difference (*p* = 0.056) (Figure 5E).

## 4. Discussion

PPIV1 is currently reported in Asia [8], Europe [13], South America [9], and the U.S. [10,11]. Sequencing data provides evidence of genetic diversity between different strains and across geographic locations. Recent phylogenetic evidence from PPIV1 F gene sequences compiled internationally [21] and supported by sequences obtained at the ISU VDL (data not shown) suggest at least two co-circulating phylogenetic clades within U.S. swine. The MN16 and IA17 PPIV1 isolates share approximately 98.2% whole virus nucleotide homology and based on the analysis presented in this study, were observed within the same phylogenetic clade. Both PPIV1 isolates were acquired when the knowledge of PPIV1 genetic diversity in the U.S. was limited. However, the MN16 and IA17 PPIV1 were the only isolates available for comparison of different strains at the time of this study based on the difficulty of isolating PPIV-1 from clinical samples, perhaps due to viral incompatibility with available cell lines or the lack of clinical samples with sufficient concentrations of virus to ensure isolation. In contrast, the nucleotide homology between the HPIV1 and both IA17 and MN16 isolates used in this study was approximately 64.9% (Figure 1). Reports have demonstrated that HPIV3 sequences exhibit even greater genetic differences from PPIV1 [8], which suggests significant genetic diversity among human and porcine parainfluenza viruses. Phylogenetic analysis across genome regions between the HPIV1 and PPIV1 isolates used in this study demonstrated greater diversity than what was observed at the whole-virus level (Table 3). Pairwise differences ranged from 29.23%–38.66% with MN16 vs. HPIV1 and 28.97–38.77% with IA17 vs. HPIV1.

In the current experimental study, there were trends suggesting minimal differences between the IA17 and MN16 PPIV1. The IA17 challenged pigs trended higher in microscopic lung lesion and IHC scores at 5 DPI (Figure 5B,E). Additionally, IHC scores in the trachea (*p* = 0.024) and NT (*p* = 0.027) were significantly increased in the IA17 challenged pigs relative to the MN16 group (Figure 5C,D). However, differences in virus shedding (Figure 3A), and replication (Figure 3B) between IA17 and MN16 were lacking. In addition, no significant differences were observed in average daily gain by 28 DPI. Similar to a previous PPIV1 study using the MN16 isolate [14], challenge with MN16 or the IA17 isolate did not cause clinical respiratory disease or result in appreciable gross pulmonary lesions. Only the IA17 had marginally increased median microscopic lung lesion composite scores relative to the Ctrl group (Figure 5B, *p* = 0.060) while no significant difference was observed between the MN16 and Ctrl groups (*p* > 0.1) suggesting microscopic lung lesions are mild despite the high level of viral replication detected in the respiratory tract (Figure 3B and Figure 5C–E). Regardless, it remains unknown if PPIV1 is a primary pathogen in the porcine respiratory disease complex based on these data or if greater genetic differences between strains of PPIV1 in the U.S, not represented in this study, may impact clinical outcomes or differences in clinical signs of respiratory disease observed between experimental studies and what is described in field infections. Nucleotide and amino acid diversity between IA17 and MN16 at the gene level were less than 5% suggesting minimal genetic diversity between the two isolates may have limited the ability to observe clinical differences in this study. Differences in virulence between PIV strains or serotypes has been documented in other species including mice [22] and humans [23]. These differences in virulence have been speculated to be caused by host factors such as age at exposure [24] and genetics [25] or viral factors due to antigenic variability between strains [26]. Additional PPIV1 isolated from swine with greater genetic diversity are needed for experimental studies to effectively evaluate pathogenesis and virulence differences that could implicate PPIV1 as a primary cause of porcine respiratory disease.

Parainfluenza viruses have been associated with clinical disease in humans, disproportionately affecting young children [27] and the immunocompromised [7,28]. A variety of clinical signs have been attributed to BPIV3 in cattle [29,30,31]. Experimental inoculation studies aimed at reproducing clinical disease have resulted in clinical signs ranging from asymptomatic infection to severe bronchopneumonia [32,33,34]. However, recent experimental evidence in cattle is lacking as most research was conducted in the 1960s and 1970s [5]. In swine, clinical disease associated with PPIV1 has not been observed under controlled experimental inoculation [14]. However, experimental results conflict with field case reports which associate PPIV1 with clinical respiratory disease in the absence of other known respiratory pathogens [15].

Based on results demonstrated in this study, the ability of HPIV1 to replicate in swine (Figure 3B) and PPIV1 antibody to cross-react with HPIV antigen using the HPIV1-3 specific ELISA (Figure 4C) suggest that PIVs could naturally cross species barriers. A BPIV3 spillover event was previously documented in pigs [35,36]. However, this virus did not become established in swine using serosurveillance conducted with wv-ELISA from 876 field sera collected 2008 to 2009. It has been suggested that antibody cross-reactive immune responses are responsible for host restriction of paramyxoviruses, although dependent on the degree of cross-protection [37]. Studies have suggested that HPIV3 and BPIV3 are limited to humans and cattle, respectively, due to preexisting natural immunity [38]. Further evidence has demonstrated that dogs vaccinated with measles, a related genus *Morbillivirus* in the Paramyxoviridae family, are protected against subsequent challenge with canine distemper [39].

The results described in this study demonstrated that HPIV1 was readily detected from inoculated pigs by RT-qPCR and supported by virus isolation at 5 DPI. However, HPIV1 was inconsistently detected in NS from 1–4 DPI suggesting that, despite replication, shedding and sustained transmission of a human PIV among pigs may be limited, at least with the strain used in the current study. Interestingly, IA17 PPIV1 antibodies were also shown to cross-react to some degree with HPIV1-3 antigen as detected by a modified wv-ELISA, which also suggests some antigenic similarities between porcine and human PIVs. However, the lack of cross-neutralization demonstrated by SVN between HPIV1 and PPIV1 strains did not correspond with the ELISA results. This is not entirely unexpected considering the HPIV1-3 assay was adapted from a commercial human ELISA kit that could detect antibodies against HPIV 1-3, while the SVN was specific to the MN16 virus antigen. Collectively, these data suggest potential cross-species transmission may occur among different strains of PIV although using swine as a model for HPIV1 infection appears limited. Additional studies are needed to further assess cross-species transmission of PIV in humans and swine and to evaluate cross-neutralization between different PPIV1 isolates as well as across different species.

## 5. Conclusions

The pathogenesis of two different but genetically related PPIV1 isolates used in this study, IA17 and MN16, resulted in minimal clinical signs, gross lesions, or microscopic lesions in swine. In addition, the results from this study indicate that HPIV1 can replicate in nursery swine and induce seroconversion without clinical signs of respiratory disease. However, more research is needed to investigate the epidemiology of different strains of PPIV1 in the U.S. to understand the pathogenesis and their potential to cause clinical disease in swine or if other strains of HPIV may infect pigs. Isolation of genetically divergent PPIV1 within different phylogenetic clades is necessary to conclude whether differences in disease severity exist between strains. In addition, it remains unclear what role other viral or bacterial coinfections may contribute to the pathogenesis of PPIV1 and its contribution to the porcine respiratory disease complex. Coinfection studies with other known swine respiratory pathogens, such as *Streptococcus suis*, *Mycoplasma hyopneumoniae*, influenza A virus, or porcine reproductive and respiratory syndrome virus, will help establish how PPIV1 contributes to swine respiratory disease and assist veterinarians and producers with management decisions.

## Figures and Tables

**Figure 1 vetsci-10-00018-f001:**
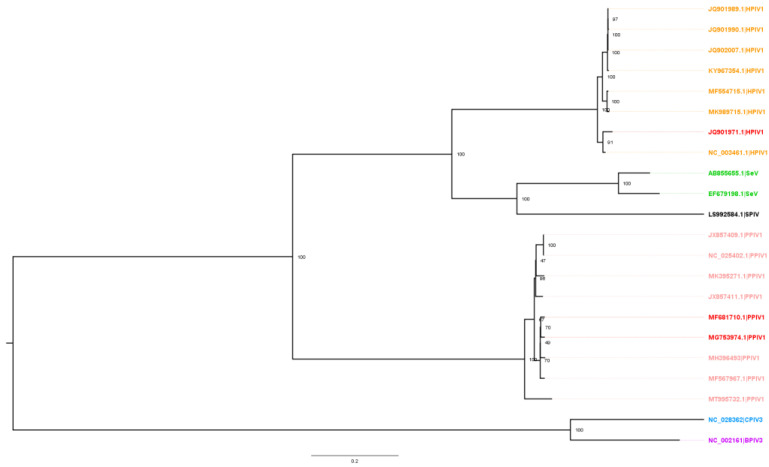
Maximum likelihood phylogenetic tree of whole-genome sequences from porcine parainfluenza virus 1 (PPIV1; pink), Sendai Virus (SeV; green), Giant squirrel parainfluenza virus (SPIV; black), caprine parainfluenza virus 3 (CPIV3; blue), bovine parainfluenza virus 3 (BPIV3; purple), and Human parainfluenza virus 1 (HPIV1; orange). Maximum likelihood bootstrap support values are shown at each node. The red sequences correspond to the isolates used for challenge including JQ901971.1 (HPIV1), MG753974.1 (IA17), and MF681710.1 (MN16).

**Figure 2 vetsci-10-00018-f002:**
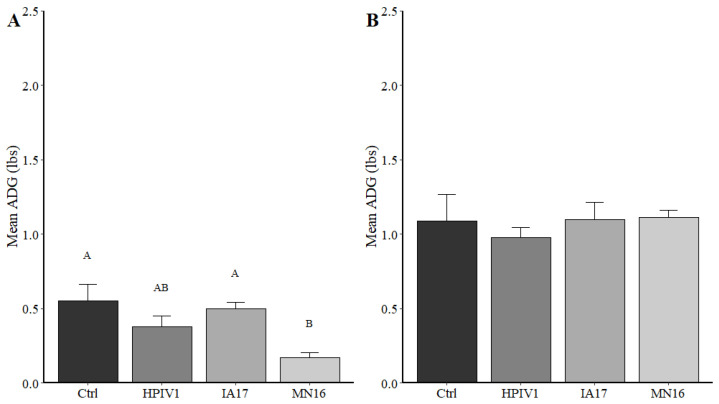
Mean average daily gain reported from the Ctrl, HPIV1, IA17 and MN16 groups. (**A**) Average daily gain measured at 5 DPI. Weight gain was significantly reduced in the MN16 group relative to the IA17 and Ctrl group. Different letters above the bar in the graph indicates a significant difference (*p* < 0.05) between experimental groups. (**B**) Average daily gain measured at 28 DPI. No significant differences were observed in average daily gain between any groups. Ctrl: Control group; HIPV1: Human parainfluenza virus 1; IA17: USA/IA84915LG/2017; MN16: USA/MN25890NS/2016.

**Figure 3 vetsci-10-00018-f003:**
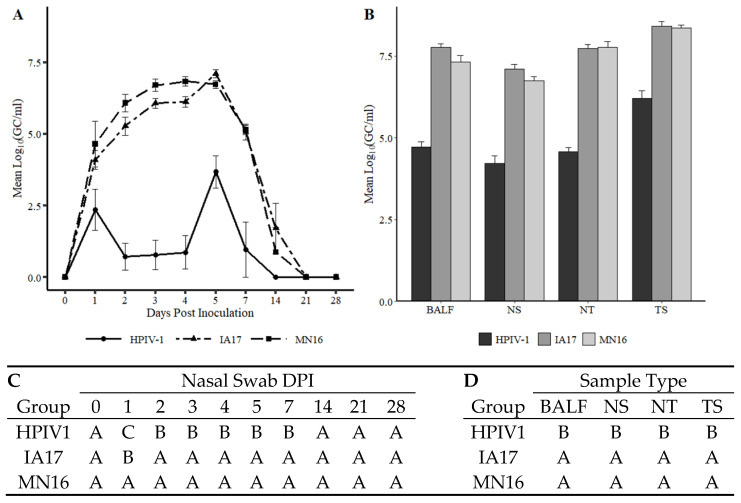
Detection of HIPV1, IA17, or MN16 viral nucleic acid from experimental specimens by RT-qPCR. (**A**) Mean HPIV1, IA17, and MN16 GC/mL detected in antemortem NS homogenate and standard error of the mean (SEM) error bars. (**B**) Mean HPIV1, IA17, and MN16 GC/mL detected in BALF, NS, NT and TS necropsy sample homogenates and SEM error bars. (**C**) Connecting letters report comparing nasal swab GC/mL at each DPI is provided to indicate significance at *p* < 0.05. Groups with different letters within a column are significantly different. A Tukey–Kramer adjustment for multiple comparisons was applied to correct for family-wise error rate. (**D**) Connecting letters report comparing BALF, NS, NT and TS GC/mL collected at necropsy is provided to indicate significance at *p* < 0.05. Groups with different letters within a column are significantly different. A Tukey–Kramer adjustment for multiple comparisons was applied to correct for family-wise error rate.

**Figure 4 vetsci-10-00018-f004:**
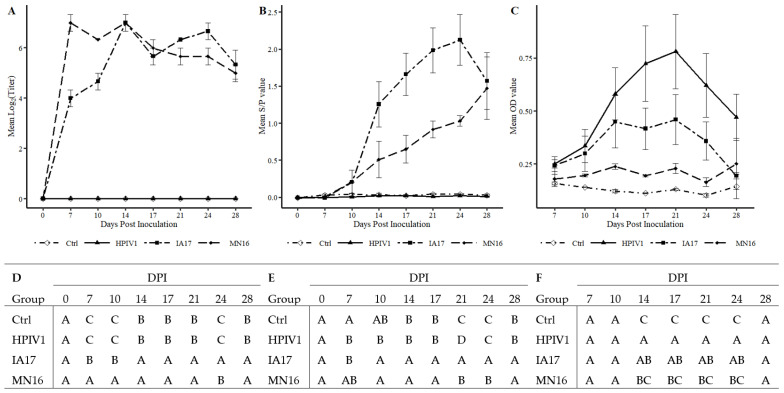
Antibody detection by SVN, PPIV1 wv-ELISA and HPIV1-3 ELISA in Ctrl, HPIV1, IA17, and MN16 serum. (**A**) Mean PPIV1 log_2_ SVN antibody titers and SEM error bars. (**B**) Mean PPIV1 wv-ELISA S/P values and SEM error bars. (**C**) Mean OD values detected in the HPIV1-3 wv-ELISA and SEM error bars. (**D**) Connecting letters report comparing mean PPIV1 log_2_ SVN antibody titers at each DPI is provided to indicate significance at *p* < 0.05. Groups with different letters within a column are significantly different. A Tukey–Kramer adjustment for multiple comparisons was applied to correct for family-wise error rate. (**E**) Connecting letters report comparing mean PPIV1 wv-ELISA S/P values at each DPI is provided to indicate significance at *p* < 0.05. Groups with different letters within a column are significantly different. A Tukey–Kramer adjustment for multiple comparisons was applied to correct for family-wise error rate. (**F**) Connecting letters report comparing mean HPIV1 OD values at each DPI is provided to indicate significance at *p* < 0.05. Groups with different letters within a column are significantly different. A Tukey–Kramer adjustment for multiple comparisons was applied to correct for family-wise error rate.

**Figure 5 vetsci-10-00018-f005:**
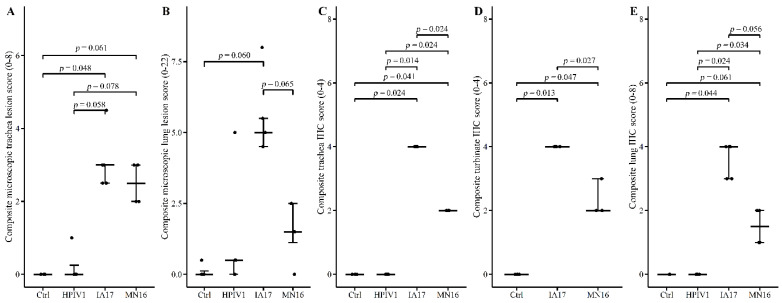
Microscopic lesion and IHC scores by treatment group at 5 DPI. Error bars represent 25th, 50th, and 75th quartiles, respectively. Points are jittered horizontally to prevent overplotting. Significance brackets are provided if *p* < 0.1. (**A**) Microscopic composite trachea lesion score. Pig 8168 from the MN16 group was missing and not included in the analysis. (**B**) Microscopic lung lesion scores. (**C**) Microscopic trachea IHC scores. (**D**) Microscopic nasal turbinate IHC scores. The HPIV group was excluded from this analysis due to missing observations. Pig 8157 from the MN16 group was missing and not included in the analysis. (**E**) Microscopic lung IHC scores.

**Figure 6 vetsci-10-00018-f006:**
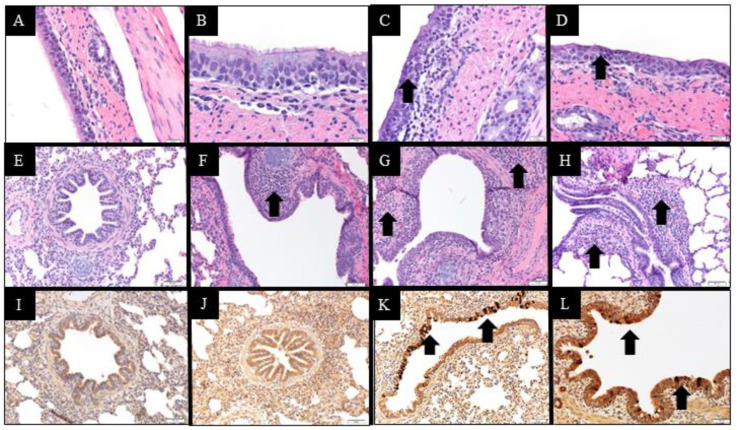
Microscopic lesions of the upper and lower respiratory tract in PPIV1 and HPIV1 challenged nursery pigs. Images A-D, E-H, and I-L correspond to H&E stain of the trachea (400×), H&E stain of the lung (200×), and IHC of the lung (200×), respectively. Images from the same column (**A**–**I**, **B**–**J**, **C**–**K** and **D**–**L**) correspond to Ctrl, HPIV1, IA17, and MN16 groups, respectively. Microscopic lesions were lacking or minimal in the trachea (**A**,**B**; score 0) and no PPIV1 signal was observed in the lung (**I**,**J**; score 0) of the Ctrl and HPIV1 groups, respectively. Lung lesions were also lacking in the Ctrl group (**E**; score 0). Mild to moderate epithelial attenuation and loss of cilia (black arrows) was observed in the IA17 (**C**; score 2.5) and MN16 (**D**; score 3.0) trachea. Mild to moderate peribronchiolar cuffing (black arrows) consisting of mononuclear cells were observed in the lung from HPIV1 (**F**; score 0.5), IA17 (**G**; score 5.0), and MN16 (**H**; score 2.5) groups. PPIV1 specific signal indicated as brown staining (black arrows) was observed in the bronchiolar epithelium in the IA17 (**K**; score 3.0) and MN16 (**L**; score 2.0) groups. PPIV-1 signal was also observed in the respiratory epithelium of the trachea and turbinate in the IA17 and MN16 groups (data not shown).

**Table 1 vetsci-10-00018-t001:** Experimental design comparing challenge with Human parainfluenza virus 1 (ATCC^®^ VR-94™), and Porcine parainfluenza virus 1 isolates USA/MN25890NS/2016 and USA/IA84915LG/2017 in conventional nursery pigs. NA: Not applicable.

Group	Virus	GenBank	N	Necropsy (N)	
				5 DPI	28 DPI
Ctrl	NA	NA	6	4	2
HPIV1	Strain C35 ATCC^®^ VR-94^™^	JQ901971	8	5	3
IA17	USA/IA84915LG/2017	MG753974	8	5	3
MN16	USA/MN25890NS/2016	MF681710	8	5	3

**Table 2 vetsci-10-00018-t002:** Table of Human parainfluenza virus 1 RT-qPCR primers and probes.

Primer NAME	Sequence (5′-3′)
HPIV1-For	TCGGTGCTGTTATTGGTACCAT
HPIV1-Rev	TCTCGTGCTTCAGCTAATGCA
HPIV1-Probe	FAM-CACTAGGAGTAGCCACAGCTGCCCAGA-Iowa Black

**Table 3 vetsci-10-00018-t003:** Gene-level pairwise % differences between MN16, IA17, and HPIV1 virus isolates used in this study.

			MN16 vs. IA17		MN16 vs. HPIV1		IA17 vs. HPIV1	
Genome Region	Gene [Length (nt), PPIV/HPIV]	Protein [Length (aa), PPIV/HPIV]	Δnt, N (%) ^§^	Δaa, N (%) ^§^	Δnt, N (%) ^§^	Δaa, N (%) ^§^	Δnt, N (%) ^§^	Δaa, N (%) ^§^
Whole Genome	15334/15516	NA	285 (1.81%)	NA	5111 (32.44%)	NA	5125 (32.53%)	NA
Nucleoprotein (NP)	1581/1575	527/525	36 (2.28%)	8 (1.53%)	466 (29.23%)	136 (25.78%)	462 (28.97%)	135 (25.59%)
Phosphoprotein (P)	1725/1803	575/601	79 (3.07%)	32 (3.93%)	978 (38.07%)	445 (54.60%)	986 (38.43%)	453 (55.58%)
Matrix (M)	1044/1047	348/349	18 (1.72%)	3 (0.87%)	319 (30.46%)	86 (24.86%)	324 (30.94%)	85 (24.57%)
Fusion (F)	1674/1668	558/556	25 (1.49%)	13 (2.35%)	579 (34.38%)	200 (35.78%)	578 (34.31%)	198 (35.40%)
Hemagglutinin /Neuraminidase (HN)	1731/1728	577/576	34 (1.91%)	26 (4.98%)	681 (38.66%)	349 (67.76%)	683 (38.77%)	347 (67.35%)
Large Polymerase (L)	6708/6693	2236/2231	86 (1.28%)	57 (2.86%)	1901 (28.33%)	962 (48.19%)	1908 (28.43%)	967 (48.44%)

§ Nucleotide (nt) and amino acid (aa) differences were calculated for each gene or protein, respectively.

## Data Availability

The data presented in this study are available on request from the corresponding author. The data are not publicly available due to pending future publications.

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
