# Peer review of "Characterization of Two Porcine Parainfluenza Virus 1 Isolates and Human Parainfluenza Virus 1 Infection in Weaned Nursery Pigs"

_vetsci, 2022, doi:10.3390/vetsci10010018_

Round 1
Reviewer 1 Report
Manuscript vetsci-2092654 by Welch et al. describes the characterization of two PPIV1 strains and one HPIV1 strain. Authors evaluate the three viruses that differ in genetic similarity, ADG, virus replication, anti-viral antibodies responses, and pathogenicity and prove that the two PPIV1 strains were associated with minor clinical signs, and HPIV1 strain shows capability of cross-species transmission.
The manuscript was well written, and highly organized. I have no significant comments on this work.
I have only some minor comments.
Introduction: it is recommended that authors include brief descriptions about PPIV virus, genome type, size, translated proteins and their biological functions so that readers would be easier to understand phylogenetic results.
Materials and Methods: it is recommended that name of reagents used here be consistent, and introduce acronym when the full term shows up the first time. e.g. in line 67, nasal swab showed up first time so that "(NS)" should be introduced here instead of line 162; in line 110, should it be biotinylated goat anti-mouse IgG the same as line 93? in line 87, 96-well plate needs describer; in line 167, PRRSV needs full term.
Section 2.3: S/P ratio calculation replies on positive control mean OD, but there is no description about how the positive control was prepared.
Section 2.10: may be I am not a stats pro, but where is the linear regression results in the paper. also why not use data standardization method by subtracting the mean and dividing by stdv, but use two different log transformation?
Figure 4: in connecting letters report, some have double letters. e.g. in 4C, strain IA17 has 'AB' at 14 DPI. what does it mean? does it mean IA17 has OD value not significantly different from both MN16 and HPIV1? I suggest to explain it in figure text.
Section 3.7: in line 395, should p value = 0.056?
Author Response
Ref.: Manuscript ID: vetsci-2092654
December 19, 2022
Dear Dr. Dashielle Li:
Thank you for considering our manuscript entitled “Characterization of two porcine parainfluenza virus 1 isolates and human parainfluenza virus 1 infection in weaned nursery pigs” for publication in Veterinary Sciences.
The reviewers indicated a series of considerations that we have addressed in the response to reviewers provided below. In addition, we have revised the manuscript extensively per the reviewer’s suggestions and to add more clarity to the results presented in the manuscript. Additionally, figure 6 was updated to provide the magnification of the images, provide the scale bars for each image, and to provide examples of lesion scores for the micrographs presented as examples of the lesions described in the tissues.
The specific changes are detailed in the response to reviewers document and in the tracked changes version of the revised manuscript. Thank you again for your time and constructive evaluation of our paper.
Sincerely,
Phillip Gauger
Iowa State University
Our responses to the reviewers are presented in italics and start with “A:”. Line numbers in reviewer comments reference the original submission; line numbers in our comments correspond to the revised manuscript with track changes.
Reviewer 1: Comments and Suggestions for Authors
Manuscript vetsci-2092654 by Welch et al. describes the characterization of two PPIV1 strains and one HPIV1 strain. Authors evaluate the three viruses that differ in genetic similarity, ADG, virus replication, anti-viral antibodies responses, and pathogenicity and prove that the two PPIV1 strains were associated with minor clinical signs, and HPIV1 strain shows capability of cross-species transmission.
The manuscript was well written, and highly organized. I have no significant comments on this work. I have only some minor comments.
Introduction: it is recommended that authors include brief descriptions about PPIV virus, genome type, size, translated proteins and their biological functions so that readers would be easier to understand phylogenetic results.
A: The author appreciates the reviewer’s suggestion to add more genetic and detailed information regarding PPIV in swine in the introduction of the manuscript. This information has been added to lines 40-45.
Materials and Methods: It is recommended that name of reagents used here be consistent, and introduce acronyms when the full term shows up the first time. e.g. in line 67, nasal swab showed up first time so that "(NS)" should be introduced here instead of line 162; in line 110, should it be biotinylated goat anti-mouse IgG the same as line 93? in line 87, 96-well plate needs describer; in line 167, PRRSV needs full term.
A: The author appreciates the reviewer’s suggestions for the acronyms. The abbreviation NS was included in line 75 considering that is the first time it is mentioned in the paper.
A: Two different immunocytochemistry assays were used to detect PPIV1 or HPIV1 in cell culture. The PPIV1 immunocytochemistry assay used a biotinylated goat anti-mouse IgG as a secondary antibody with a third streptavidin-HRP conjugate to help amplify the signal. The HPIV1 immunocytochemistry assay used an antibody that was obtained commercially and an additional amplification step was not needed. Additional details regarding the immunocytochemistry assays were added to lines 99-110 and 118-123.
A: The virus titrations described in line 95 used 96 well Microtiter plates from Thermo Fisher Scientific. This information was added to lines 84-85 and again at lines 95.
A: Porcine reproductive and respiratory syndrome virus was added to line 195.
Section 2.3: S/P ratio calculation replies on positive control mean OD, but there is no description about how the positive control was prepared.
A: The wv-ELISA was published in detail in ‘Welch M, Krueger K, Zhang J, Piñeyro P, Magtoto R, Wang C, Giménez-Lirola L, Strait E, Mogler M, Gauger P. Detection of porcine parainfluenza virus type-1 antibody in swine serum using whole-virus ELISA, indirect fluorescence antibody and virus neutralizing assays. BMC Vet Res. 2022 Mar 21;18(1):110. doi: 10.1186/s12917-022-03196-6. PMID: 35313864; PMCID: PMC8935814.’ The calculation of the S/P ratio is described in that manuscript. The corresponding positive serum sample that was collected from an experimental PPIV1 inoculated pig at DPI 21 and appropriate negative controls collected from a PPIV1 negative pig were tested in duplicate within each wv-ELISA plate to normalize OD results and calculate the S/P ratios. This was added to the manuscript in lines 162-165.
Section 2.10: Maybe I am not a stats pro, but where is the linear regression results in the paper. also why not use data standardization method by subtracting the mean and dividing by stdv, but use two different log transformation?
A: The results from the linear models are provided as connecting letters reports below the figures (figure 4D-F). The author is unsure whether Reviewer 1 is requesting specific model coefficients or lsmeans. P-values are provided from pairwise comparisons in the manuscript text. The S/P and OD values were log transformed to improve the fit of the model based on assessment of residual diagnostic plots provided in PROC MIXED. Identical p-values will be obtained regardless of whether a natural logarithm or log10 transformation is used as long as the data are logarithmically transformed. For genomic copies, a log10 transformation was used as the author considered the statistical output more intuitive prior to back transformation.
Figure 4: in connecting letters report, some have double letters. e.g. in 4C, strain IA17 has 'AB' at 14 DPI. what does it mean? does it mean IA17 has OD value not significantly different from both MN16 and HPIV1? I suggest to explain it in figure text.
A: The figure 4 legend was updated to add additional descriptions of the connecting letters report where it states “groups with different letters within a column are significantly different.” Conversely, groups that share letters show a lack of significant differences. In the example provided by the reviewer, IA17 differs significantly from the Ctrl but not HPIV1 and MN16. The manuscript was updated at lines 353, 356, 387, 389, and 392 (see tracked changes).
Section 3.7: in line 395, should p value = 0.056?
A: The author appreciates the reviewer noticing this typing error and the p-value should be 0.056 between MN16 and IA17 IHC scores in the lung. The text was updated at line 445.

Reviewer 2 Report
In this Ms, authors performed a systemic investigation including epidemiology, phylogenetic analysis, pathogenesis and immunology of currently circulating two PPIV1 isolates, although there are only a few strains available. Of note, this study provides first evidence of potential pathogenesis differences between the two strains. Additionally, a human parainfluenza virus (HPIV1) was included to evaluate in swine model for potential cross-species transmission.
Overall, both the experiment design and writing are very well, and authors conducted a comprehensive study on the two PPIV1 isolates, and tried to replicate the clinical symptoms after inoculations of the viruses in lab conditions. Therefore, this is an excellent study on porcine parainfluenza virus 1.
I have only a few minor concerns listed as below for authors.
In Fig.4C antibody detection by HPIV1-3, I suggest authors may use Mean S/P values for Y-axis, otherwise explain.
In line 358, authors should point out the expanded macrophage & lymphocytes in submucosa in fig 6.
In line 408-410, authors might explain why it is difficult to isolate PPIV-1 from clinical sample?
In line 437, authors may consider replacing “the gene level” to “the genome level”
Author Response
Ref.: Manuscript ID: vetsci-2092654
December 19, 2022
Dear Dr. Dashielle Li:
Thank you for considering our manuscript entitled “Characterization of two porcine parainfluenza virus 1 isolates and human parainfluenza virus 1 infection in weaned nursery pigs” for publication in Veterinary Sciences.
The reviewers indicated a series of considerations that we have addressed in the response to reviewers provided below. In addition, we have revised the manuscript extensively per the reviewer’s suggestions and to add more clarity to the results presented in the manuscript. Additionally, figure 6 was updated to provide the magnification of the images, provide the scale bars for each image, and to provide examples of lesion scores for the micrographs presented as examples of the lesions described in the tissues.
The specific changes are detailed in the response to reviewers document and in the tracked changes version of the revised manuscript. Thank you again for your time and constructive evaluation of our paper.
Sincerely,
Phillip Gauger
Iowa State University
Our responses to the reviewers are presented in italics and start with “A:”. Line numbers in reviewer comments reference the original submission; line numbers in our comments correspond to the revised manuscript with track changes.
Reviewer 2: Comments and Suggestions for Authors
In this Ms, authors performed a systemic investigation including epidemiology, phylogenetic analysis, pathogenesis and immunology of currently circulating two PPIV1 isolates, although there are only a few strains available. Of note, this study provides first evidence of potential pathogenesis differences between the two strains. Additionally, a human parainfluenza virus (HPIV1) was included to evaluate in swine model for potential cross-species transmission.
Overall, both the experiment design and writing are very well, and authors conducted a comprehensive study on the two PPIV1 isolates, and tried to replicate the clinical symptoms after inoculations of the viruses in lab conditions. Therefore, this is an excellent study on porcine parainfluenza virus 1.
I have only a few minor concerns listed as below for authors.
In Fig.4C antibody detection by HPIV1-3, I suggest authors may use Mean S/P values for Y-axis, otherwise explain.
A: The HPIV1-3 ELISA was adapted to the swine sera from a kit that tests for antibody from human sera. Because of the novelty of this work, the authors did not have access to samples from other HPIV1 infected pigs outside of this study to allow us to calculate appropriate S/P cutoff values. Therefore, similar to other manuscripts that report ELISA results, we used only the optical density as the parameter to compare antibody levels between experimental groups (see as an example figure 5 in manuscript Wymore Brand M, et al., Bivalent hemagglutinin and neuraminidase influenza replicon particle vaccines protect pigs against influenza a virus without causing vaccine associated enhanced respiratory disease. Vaccine. 2022 Sep 9;40(38):5569-5578. doi: 10.1016/j.vaccine.2022.07.042. Epub 2022 Aug 18. PMID: 35987871). All the sera for this study were run on the same plate to minimize plate-to-plate variation and facilitate comparisons between groups. The purpose of the assay was to simply demonstrate that HPIV1 inoculated pigs were actually infected and seroconverted, further supporting evidence of viral replication.
In line 358, authors should point out the expanded macrophage & lymphocytes in submucosa in fig 6.
A: Figure 6 was extensively updated per the reviewer’s suggestions. Lesion scores were added to figure 6 for each of the micrograph pictures where applicable and black arrows were used to identify trachea epithelial attenuation and loss of cilia and lung lesions such as peribronchiolar cuffing from the expansion of monocytes and lymphocytes. Please see figure 6F, 6G, 6H that highlight the macrophages and lymphocytes surrounding airways in the lung.
In line 408-410, authors might explain why it is difficult to isolate PPIV-1 from clinical sample?
A: This comment was inserted from lines 457-461.
In line 437, authors may consider replacing “the gene level” to the “genome level”
A: “Gene level” was changed to “across genome regions” at lines 466-467 (see tracked changes).

Reviewer 3 Report
A manuscript (vetsci-2092654) entitled “Characterization of two porcine parainfluenza virus 1 isolates and human parainfluenza virus 1 infection in weaned nursery pigs” by M. Welch et al is mainly describing about the experimental infection of two porcine respirovirus 1 isolates obtained at 2016 and 2017. Though those two isolates did not cause the significant respiratory disease in pigs, this study provides an important knowledge for the infectious diseases of pigs and it is worth to be noted in Veterinary Sciences.
However, before accepting this manuscript, this reviewer would like to tell the authors 3 major comments and 24 minor comments and suggestions, those of which should be clarified by the authors. Thus, this reviewer recommends a major revision of this manuscript.
Major comments;
1. As described in Introduction section, porcine parainfluenza virus 1, Sendai virus, human parainfluenza virus 1, bovine parainfluenza virus 3, caprine parainfluenza virus 3, and giant squirrel parainfluenza virus are classified all at once into the genus Respirovirus. In addition, full genome sequence data of viral strains belong to each species are all available from DNA sequence data base. As the phylogenetic tree drawn in Figure 1 contains only porcine parainfluenza virus 1, Sendai virus, giant squirrel parainfluenza virus human parainfluenza virus 1, but not contains bovine parainfluenza virus 3 nor caprine parainfluenza virus 3. It is requested to draw the maximum likelihood phylogenetic tree containing at least one strain from each species in the genus Respirovirus.
2. Full genome sequence data of two porcine parainfluenza virus 1 isolates are used to draw the phylogenetic tree, but no description how author obtained the sequence results from two strains, USA/MN25890NS/2016 (MN16, passage 9) and USA/IA84915LG/2017 68 (IA17, passage 5), was found in the manuscript. Please add the subsection in Materials and Methods, and describe the methods if the genome sequences were authors’ original work. If not, please add some proper information regarding to the genome sequence of two strains.
3. This manuscript has so many abbreviated technical terms, such as nasal turbinate (NT, line 56), bronchoalveolar lavage fluid (BALF, line 102), room temperature (RT, line 131), serum virus neutralization (SV, line 149), immunocytochemistry (ICC, line 157), nasal swabs (NS, line 162), average daily gain (ADG, line 180), tracheal mucosal swabs (TS, line 182), lobes of lung (LG, line 183), genomic copies per ml (GC/ml, line 211) and Necropsy (N, Table 1). It is hard for the readers including this reviewer, who are not familiar to use above abbreviations, to read this manuscript. It is also confusing to use same abbreviation such as RT (room temperature) and RT (reverse transcription)-qPCR, N (necropsy) and N (nucleoprotein). In addition, “PRRSV” (line 167) and “IHC scores” (line 222) are used without the spelled-out form. Please try to decease the use of abbreviation forms, if they are not necessary.
Minor comments;
1. Lines 26, Abstract section; “control (Ctrl) group” had better be described as “non-inoculated negative control (Ctrl) group”
2. Lines 26, Abstract section; “MN16 and Ctrl” had better be described as “MN16 and Ctrl groups”
3. Lines 67-71, Materials and Methods; Please show which cell line was used to establish PPIV1 USA/MN25890NS/2016 strain (MN16, passage 9) and USA/IA84915LG/2017 68 strain (IA17, passage 5). Was LLCK-MK2 cell line used as same as human parainfluenza virus 1? Please clarify.
4. Lines 92, Materials and Methods; Please clarify the target protein to which the monoclonal antibody reacts.
5. Lines 104, 109,111 and 155, Materials and Methods; “2x”, “3x” and “4x” had better be shown as “2 times”, “3 times” and “4 times” respectively.
6. Lines 105 and 155, Materials and Methods; “2hrs” should be shown as “2h” to uniform the description in the manuscript.
7. Lines123-124, Materials and Methods; A sentence “500 mL of the MN16 isolate was grown in cell culture as described above.” had better be described as “500 mL of the MN16 isolate solution was prepared in cell culture as described above.”
8. Lines124-125, Materials and Methods; A meaning of sentence “The clarified supernatant was ultracentrifuged 124 at 140,992xg for 3h and washed twice with phosphate buffered saline” is not clear. Did the authors precipitate the virus by centrifugation and washed the pellet two times by PBS? If so, how many volumes of PBS was used to resuspend the virus? Please clarify the methods. In addition to above, did the authors use the virus solution for the ELISA antigen without the inactivation procedure? Please clarify this point as well.
9. Lines 128,192 and 193, Materials and Methods; “xx% w/v” had better be shown as “xx% (w/v)”.
10. Lines 152-153, Materials and Methods; “a concentration of 4000 50% tissue culture infectious dose per ml (TCID50/ml) (200 TCID50/50μl)” had better be shown as “a concentration of 4000 50% tissue culture infectious dose (TCID50) per ml (200 TCID50/50μl)”
11. Line 163, Materials and Methods; Is “RT-rtPCR” correct? It is described in line 203 as “PPIV1 RNA was detected by quantitative reverse transcription PCR (RT-qPCR)”. Please unify the description and delete the redundancy from the Materials and Methods section, if they mean same procedure.
12. Lines 169-170, Materials and Methods; A sentence “Six pigs were assigned to the control (Ctrl) group” had better be described as “Six pigs were assigned to the non-inoculating negative control (Ctrl) group”
13. Lines 187-188, Materials and Methods; A definition of “respiratory scores (0: normal, 1: mild, 2: moderate, 3: severe)” is not clear. Please show how grading the damage of tissues, like damaged area%, or degree of damage.
14. Line 216, Materials and Methods; “(Applied Biosystems, Foster City CA)” is no more required, because a 7500 Fast thermocycler is already shown in above line 209.
15. Table 2; A HPIV1-PROBE contains a quencher at 3’ end, but not contain fluorescent dye at 5’ end. Please confirm if the sequence is properly shown or not.
16. Line 228-235, Materials and Methods; Criteria for grading the lesion scores are not clear. How did the authors determine the degree of tissue lesions of peribronchiolar cuffing (0-4), airway epithelial necrosis (0-4), suppurative bronchiolitis (0-4), epithelial microabscesses (0-3), interstitial pneumonia (0-4), alveolar edema (0-3), trachea epithelial necrosis (0-4) and tracheitis (0-4)? Please clarify.
17. Line 259, Results; “SeV/SPIV” should be “SeV, SPIV”.
18. Line 268, Results; “PVC polyprotein” should be “P protein”, because no amino-acid sequence comparison of C nor V protein in P gene is shown in Table 3.
19. Figure 2; Sample explanations outside the figure 2A and 2B are not required, because the name of each sample is already shown in X axis of figure 2A and 2B.
20. Figure 3; There are no explanation in the caption about the table shown by “A”, “B” and “C” located under the figure 3A and 3B.
21. Figure 4; There are no explanation in the caption about the table shown by “A”, “AB”, “B” and “C” located under the figure 4A, 4B and 4C. Please show the cut off OD value of HPIV1-3 commercial ELISA kit.
22. Figure 6; Relations between microscopic lesions and lesion scores are not shown clearly. Correlations between the scores of Figure 5 and lesions shown in Figure 6 are not explained clearly. For example, which score did the authors give for tissue samples showing mild to moderate peribronchiolar cuffing consisting of mononuclear cells in figure (F), (G), and (H). In addition to above, one cannot understand which color is PPIV1 specific signal. Please consider to add the explanation like the dark brown colored specific signals or add the arrows indicating the antigen specific signal. Moreover, magnitudes of microscopic images are not shown. Please consider to add the scale bar in each histopathological photograph.
23. Line 400, Discussion; “Porcine parainfluenza virus 1” should be PPIV1.
24. Lines 403-404, Discussion; If two phylogenetic clades are found in the F gene of PPIV1, please consider to draw the F gene based phylogenetic tree containing at least one strain from each species in the genus Respirovirus, and show to which clade PPIV1 MN16 and IA17 belong.
Author Response
Ref.: Manuscript ID: vetsci-2092654
December 19, 2022
Dear Dr. Dashielle Li:
Thank you for considering our manuscript entitled “Characterization of two porcine parainfluenza virus 1 isolates and human parainfluenza virus 1 infection in weaned nursery pigs” for publication in Veterinary Sciences.
The reviewers indicated a series of considerations that we have addressed in the response to reviewers provided below. In addition, we have revised the manuscript extensively per the reviewer’s suggestions and to add more clarity to the results presented in the manuscript. Additionally, figure 6 was updated to provide the magnification of the images, provide the scale bars for each image, and to provide examples of lesion scores for the micrographs presented as examples of the lesions described in the tissues.
The specific changes are detailed in the response to reviewers document and in the tracked changes version of the revised manuscript. Thank you again for your time and constructive evaluation of our paper.
Sincerely,
Phillip Gauger
Iowa State University
Our responses to the reviewers are presented in italics and start with “A:”. Line numbers in reviewer comments reference the original submission; line numbers in our comments correspond to the revised manuscript with track changes.
Reviewer 3: Comments and Suggestions for Authors
A manuscript (vetsci-2092654) entitled “Characterization of two porcine parainfluenza virus 1 isolates and human parainfluenza virus 1 infection in weaned nursery pigs” by M. Welch et al is mainly describing about the experimental infection of two porcine respirovirus 1 isolates obtained at 2016 and 2017. Though those two isolates did not cause the significant respiratory disease in pigs, this study provides an important knowledge for the infectious diseases of pigs and it is worth to be noted in Veterinary Sciences.
However, before accepting this manuscript, this reviewer would like to tell the authors 3 major comments and 24 minor comments and suggestions, those of which should be clarified by the authors. Thus, this reviewer recommends a major revision of this manuscript.
Major comments;
- 1. As described in Introduction section, porcine parainfluenza virus 1, Sendai virus, human parainfluenza virus 1, bovine parainfluenza virus 3, caprine parainfluenza virus 3, and giant squirrel parainfluenza virus are classified all at once into the genus In addition, full genome sequence data of viral strains belong to each species are all available from DNA sequence data base. As the phylogenetic tree drawn in Figure 1 contains only porcine parainfluenza virus 1, Sendai virus, giant squirrel parainfluenza virus human parainfluenza virus 1, but not contains bovine parainfluenza virus 3 nor caprine parainfluenza virus 3. It is requested to draw the maximum likelihood phylogenetic tree containing at least one strain from each species in the genus Respirovirus.
A: Bovine parainfluenza virus 3 (NC002161) and caprine parainfluenza virus 3 (NC028362) were selected from GenBank based on Palinski et al., 2016 and Li et al., 2020 (JS2013 strain), respectively. These were added to the MAFFT alignment and maximum likelihood phylogenetic tree per the reviewer’s suggestion. The BPIV3 and CPIV3 information was already mentioned in the manuscript at line 47 and an updated phylogenetic tree was included for Figure 1.
- 2. Full genome sequence data of two porcine parainfluenza virus 1 isolates are used to draw the phylogenetic tree, but no description how author obtained the sequence results from two strains, USA/MN25890NS/2016 (MN16, passage 9) and USA/IA84915LG/2017 68 (IA17, passage 5), was found in the manuscript. Please add the subsection in Materials and Methods, and describe the methods if the genome sequences were authors’ original work. If not, please add some proper information regarding to the genome sequence of two strains.
A: The author appreciates the attention brought to this omission from the manuscript. The whole genome sequencing of the MN16 and IA17 isolates was described in Park JY, Welch MW, Harmon KM, Zhang J, Piñeyro PE, Li G, Hause BM, Gauger PC. Detection, isolation, and in vitro characterization of porcine parainfluenza virus type 1 isolated from respiratory diagnostic specimens in swine. Vet Microbiol. 2019 Jan;228:219-225. doi: 10.1016/j.vetmic.2018.12.002. Epub 2018 Dec 5. PMID: 30593371. Table 1 provides the GenBank accession numbers. The whole genome sequencing information was added to lines 125-130.
- 3. This manuscript has so many abbreviated technical terms, such as nasal turbinate (NT, line 56), bronchoalveolar lavage fluid (BALF, line 102), room temperature (RT, line 131), serum virus neutralization (SV, line 149), immunocytochemistry (ICC, line 157), nasal swabs (NS, line 162), average daily gain (ADG, line 180),tracheal mucosal swabs (TS, line 182), lobes of lung (LG, line 183), genomic copies per ml (GC/ml, line 211) and Necropsy (N, Table 1). It is hard for the readers including this reviewer, who are not familiar to use above abbreviations, to read this manuscript. It is also confusing to use same abbreviation such as RT (room temperature) and RT (reverse transcription)-qPCR, N (necropsy) and N (nucleoprotein). In addition, “PRRSV” (line 167) and “IHC scores” (line 222) are used without the spelled-out form. Please try to decease the use of abbreviation forms, if they are not necessary.
A: The author understands the reviewer’s concerns regarding the number of acronyms used in the manuscript. The study described in the manuscript used several sample types to conduct a thorough investigation of the pathogenesis of the MN16 and IA17 PPIV1. Therefore, several acronyms were used to reduce the redundancy of using full names of sample types and test names. However, the author attempted to reduce the number of acronyms throughout the manuscript where possible. These changes are noted as “tracked changes” in the manuscript.
Minor comments:
- 1. Lines 26, Abstract section; “control (Ctrl) group” had better be described as “non-inoculated negative control (Ctrl) group”
A: The manuscript was updated per the suggestion of the reviewer and states “non-challenged negative control (Ctrl) group” at line 26.
- Lines 26, Abstract section; “MN16 and Ctrl” had better be described as “MN16 and Ctrl groups”
The manuscript was updated per the suggestion of the reviewer and states “MN16 and Ctrl groups” at line 27.
- Lines 67-71, Materials and Methods; Please show which cell line was used to establish PPIV1 USA/MN25890NS/2016 strain (MN16, passage 9) and USA/IA84915LG/2017 68 strain (IA17, passage 5). Was LLCK-MK2 cell line used as same as human parainfluenza virus 1? Please clarify.
A: The author specified in the submitted version of the manuscript that all viruses, PPIV and HPIV, were passaged in LLC-MK2 cells. The manuscript was updated to read “All parainfluenza viruses used in this study, including the PPIV1 isolate USA/MN25890NS/2016 (MN16, passage 9), isolated in 2016 from nasal swabs collected from suckling pigs with respiratory disease, USA/IA84915LG/2017 (IA17, passage 5), isolated in 2017 from the lung of nursery pigs with coughing, and the HPIV1 isolate (Strain C35 ATCC® VR-94™) were passaged in Macaca mulatta kidney cells (LLC-MK2, ATCC® CCL-7™) as previously described [9].” at lines 74-79.
- Lines 92, Materials and Methods; Please clarify the target protein to which the monoclonal antibody reacts.
A: The monoclonal antibody generation and selection was described in Park et al., 2019 (Vet Micro). The monoclonal antibody was generated to purified whole virus of MN16 PPIV1. The monoclonal antibody was selected through a screening process with a Cell-ELISA using LLC-MK2 PPIV1 infected cells. Twenty-nine monoclonal antibodies were screened and one selected for use in diagnostic assays (Mab 34A). Therefore, this monoclonal antibody was generated from one hybridoma although it targets the whole virus. Additional information was added to lines 101-102.
- Lines 104, 109, 111 and 155, Materials and Methods; “2x”, “3x” and “4x” had better be shown as “2 times”, “3 times” and “4 times” respectively.
A: The author updated the “2x”, “3x” and “4x” to “2 times”, “3 times” and “4 times” throughout the manuscript as suggested by the reviewer. Changes are recognized as “tracked changes” throughout the manuscript.
- Lines 105 and 155, Materials and Methods; “2hrs” should be shown as “2h” to uniform the description in the manuscript.
A: The author updated the “2hr” or “48-72hr” to “2h” and “48-72h” throughout the manuscript per the suggestion of the reviewer. Please note that tracked changes were used to update these specific changes.
- Lines123-124, Materials and Methods; A sentence “500 mL of the MN16 isolate was grown in cell culture as described above.” had better be described as “500 mL of the MN16 isolate solution was prepared in cell culture as described above.”
A: The author updated the manuscript per the suggestion of the reviewer at lines 142-144.
- Lines124-125, Materials and Methods; A meaning of sentence “The clarified supernatant was ultracentrifuged 124 at 140,992xg for 3h and washed twice with phosphate buffered saline” is not clear. Did the authors precipitate the virus by centrifugation and washed the pellet two times by PBS? If so, how many volumes of PBS was used to resuspend the virus? Please clarify the methods. In addition to above, did the authors use the virus solution for the ELISA antigen without the inactivation procedure? Please clarify this point as well.
A: The author appreciates the questions regarding the wv-ELISA and suggestions to clarify the methods. This assay was published in Welch M, Krueger K, Zhang J, Piñeyro P, Magtoto R, Wang C, Giménez-Lirola L, Strait E, Mogler M, Gauger P. Detection of porcine parainfluenza virus type-1 antibody in swine serum using whole-virus ELISA, indirect fluorescence antibody and virus neutralizing assays. BMC Vet Res. 2022 Mar 21;18(1):110. doi: 10.1186/s12917-022-03196-6. PMID: 35313864; PMCID: PMC8935814.
A: The whole virus used as antigen for coating wv-ELISA plates was precipitated by ultracentrifugation and the pellet washed twice with 50 mL of PBS and the final pellet was reconstituted with 100 µL of PBS. This information was added to the manuscript at lines 144-147.
A: The authors did not inactivate the MN16 virus used as antigen for the wv-ELISA. The manuscript was updated to include that “500 mL of viable MN16 isolate solution was prepared in cell culture. . . .”. This update is located at line 142.
- Lines 128, 192 and 193, Materials and Methods; “xx% w/v” had better be shown as “xx% (w/v)”.
A: The author updated the manuscript per the suggestion of the reviewer to “(w/v)”. These updates are included as tracked changes throughout the manuscript.
- Lines 152-153, Materials and Methods; “a concentration of 4000 50% tissue culture infectious dose per ml (TCID50/ml) (200 TCID50/50μl)” had better be shown as “a concentration of 4000 50% tissue culture infectious dose (TCID50) per ml (200 TCID50/50μl)”
A: The author updated the manuscript per the suggestion of the reviewer. This was updated at line 181.
- Line 163, Materials and Methods; Is “RT-rtPCR” correct? It is described in line 203 as “PPIV1 RNA was detected by quantitative reverse transcription PCR (RT-qPCR)”. Please unify the description and delete the redundancy from the Materials and Methods section, if they mean same procedure.
A: The author appreciates the comment by the reviewer. The use of RT-rtPCR is correct in this particular description of the animal study design. During prescreening, sows and pigs were evaluated for PPIV1 using a reverse transcription real-time PCR (RT-rtPCR). The use of a true quantitative assay that determined genomic copies per mL (GC/ml) was not necessary for this phase of the pig study. RT-rtPCR is one of the accepted acronyms by the American Association of Veterinary Laboratory Diagnosticians (AAVLD). However, when using a quantitative PCR that requires a standard curve to provide genomic copies per mL, the accepted acronym is reverse transcription quantitative PCR (RT-qPCR). During the pig study, we wanted to quantify PPIV1 post-challenge and used the term RT-qPCR for any PCR that was truly quantitative.
- Lines 169-170, Materials and Methods;A sentence “Six pigs were assigned to the control (Ctrl) group” had better be described as “Six pigs were assigned to the non-inoculating negative control (Ctrl) group”
A: The author updated the manuscript to include “non-challenged negative control (Ctrl) group” at line 198.
- Lines 187-188, Materials and Methods; A definition of “respiratory scores (0: normal, 1: mild, 2: moderate, 3: severe)” is not clear. Please show how grading the damage of tissues, like damaged area %, or degree of damage.
A: The respiratory score measures the magnitude of clinical signs associated with respiratory disease in the pigs. The criteria for scoring the respiratory score was updated in the manuscript in the text at lines 215-219. However, the grading of macroscopic lung lesions, based on percent of cranioventral consolidation, is different from respiratory scores and is described in the reference provided in the manuscript at line 257 (Halbur reference). This description is provided in ‘2.9 Macroscopic and microscopic lesions and IHC scores’. This portion of the manuscript was updated to include “The weighted proportions of affected lung lobes based on percent of cranioventral consolidation were added relative to the lung volume as previously described for PRRSV [17] and previous PPIV1 pathogenesis studies [14].” at lines 254-258.
- Line 216, Materials and Methods; “(Applied Biosystems, Foster City CA)” is no more required, because a 7500 Fast thermocycler is already shown in above line 209.
A: Per the reviewer’s suggestion, this was removed from line 216 of the original manuscript.
- Table 2; A HPIV1-PROBE contains a quencher at 3’ end, but not contain fluorescent dye at 5’ end. Please confirm if the sequence is properly shown or not.
A: Table 2 was updated to include the 5’ fluorescent dye that was used in the RT-qPCR for HPIV1.
- Line 228-235, Materials and Methods;Criteria for grading the lesion scores are not clear. How did the authors determine the degree of tissue lesions of peribronchiolar cuffing (0-4), airway epithelial necrosis (0-4), suppurative bronchiolitis (0-4), epithelial microabscesses (0-3), interstitial pneumonia (0-4), alveolar edema (0-3), trachea epithelial necrosis (0-4) and tracheitis (0-4)? Please clarify.
A: The description of grading the lung, trachea and nasal turbinate microscopic lesions is described extensively in our previous PPIV1 pathogenesis study at reference: Welch M, Park J, Harmon K, Zhang J, Piñeyro P, Giménez-Lirola L, Zhang M, Wang C, Patterson A, Gauger PC. Pathogenesis of a novel porcine parainfluenza virus type 1 isolate in conventional and colostrum deprived/caesarean derived pigs. Virology. 2021 Nov;563:88-97. doi: 10.1016/j.virol.2021.08.015. Epub 2021 Sep 3. PMID: 34500147. The author updated the text with additional information and the Welch reference that includes our microscopic lesion scoring parameters “. . . . . veterinary pathologist blinded to treatment groups and based on previous PPIV1 pathogenesis studies (14).” at lines 258-2261. However, it would be redundant to include all of the scoring parameters again in this manuscript when published in this reference “Welch et al., 2021” (14).
- Line 259, Results; “SeV/SPIV” should be “SeV, SPIV”.
A: This was updated per the suggestion of the reviewer at line 292.
- Line 268, Results; “PVC polyprotein” should be “P protein”, because no amino-acid sequence comparison of C nor V protein in Pgene is shown in Table 3.
A: This was updated per the suggestion of the reviewer at line 301.
- Figure 2; Sample explanations outside the figure 2A and 2B are not required, because the name of each sample is already shown in X axis of figure 2A and 2B.
A: Per the suggestion of the reviewer, the figure legend was removed from figure 2 due to the fact the experimental groups are already included in the X axis of figure 2A and 2B.
- Figure 3; There are no explanation in the caption about the table shown by “A”, “B” and “C” located under the figure 3A and 3B.
A: Figure 3 has been updated per the reviewer’s suggestion. The 2 connecting lines reports that are presented in a table below their respective graphs are presented to show more clearly where significant differences occurred between the 3 experimental challenge groups. These connecting lines reports were given a new designation of “C” and “D”, respectively, in the figure. The connecting letters report is demonstrating a significant difference between values within a column when letters are different (A vs. B vs. C) depending on results of the statistics. This was explained with the comment in Figure 3 legend “Groups with different letters within a column are significantly different.” Please note the changes as tracked changes in this section of the manuscript.
- Figure 4; There are no explanation in the caption about the table shown by “A”, “AB”, “B” and “C” located under the figure 4A, 4B and 4C. Please show the cut off OD value of HPIV1-3 commercial ELISA kit.
A: Figure 4 has been updated per the reviewer’s suggestion. The 3 connecting lines reports that are presented in a table below their respective graphs are presented to show more clearly where significant differences occurred between the 4 experimental challenge groups These connecting lines reports were given a new designation of “D”, “E”, and “F”, respectively, in the figure. The connecting letters report is demonstrating a significant difference between values within a column when letters are different (A vs. AB vs. B vs. C) depending on results of the statistics. This was explained with the comment in Figure 4 legend “Groups with different letters within a column are significantly different.” Please note the changes as tracked changes in the manuscript.
A: The HPIV1-3 ELISA was adapted to the swine sera from a kit that tests for antibody from human sera. There were no cut-off values established for the HPIV1-3 ELISA for swine serum used in this study as this was outside of the scope of the study. The HPIV1-3 ELISA was used to compared OD values between the 4 experimental groups. The mean OD value of each group was subsequently used for statistical comparisons at each DPI. The author wanted to use the HPIV1-3 ELISA only to demonstrate the pigs challenged with HPIV1 seroconverted during the study, which confirms the pigs were infected with the virus. The HPIV1 challenged pigs demonstrated OD values significantly higher compared to the Ctrl group at 14-24 DPI indicating the pigs seroconverted.
- Figure 6; Relations between microscopic lesions and lesion scores are not shown clearly. Correlations between the scores of Figure 5 and lesions shown in Figure 6 are not explained clearly. For example, which score did the authors give for tissue samples showing mild to moderate peribronchiolar cuffing consisting of mononuclear cells in figure (F), (G), and (H). In addition to above, one cannot understand which color is PPIV1 specific signal. Please consider to add the explanation like the dark brown colored specific signals or add the arrows indicating the antigen specific signal. Moreover, magnitudes of microscopic images are not shown. Please consider to add the scale bar in each histopathological photograph.
A: Figure 6 was extensively updated per the reviewer’s suggestions. The scoring for microscopic lesions were described in the materials and methods section of the manuscript. For all tissues with lesions or IHC scores, a composite score was computed and the experimental group median score was used for statistical analysis. The lesion scores were added to figure 6 for each of the micrograph pictures where applicable and black arrows were used to identify trachea epithelial attenuation and loss of cilia and lung lesions such as peribronchiolar cuffing. The PPIV1 brown stain in the IHC micrographs was also indicated with black arrows. Pictures were updated to include a scale bar and the magnification was added to the figure legend for each tissue type.
- Line 400, Discussion; “Porcine parainfluenza virus 1” should be PPIV1.
A: The author is under the impression that acronyms are not appropriate to use at the start of a sentence. It is expected to use the entire name if starting a sentence. Therefore, the term Porcine parainfluenza virus 1 was not changed at line 449 in the manuscript because it is used at the start of the sentence.
- Lines 403-404, Discussion; If two phylogenetic clades are found in the Fgene of PPIV1, please consider to draw the F gene based phylogenetic tree containing at least one strain from each species in the genus Respirovirus, and show to which clade PPIV1 MN16 and IA17 belong.
A: The author feels the additional F gene phylogenetic tree, as a figure in this manuscript, is not necessary based on the description provided in the text of the discussion at lines 451-456. A recent manuscript [reference 21] Stadejek et al., 2022 has already been published that demonstrates 2 different lineages of PPIV1 circulating in swine worldwide with evidence that PPIV1 F gene sequences detected from swine in the United states form 2 separate monophyletic clades. Because PPIV1 is an emerging virus in swine in the US, clade names or designations have not been officially determined. However, at lines 454-455 we mention that MN16 and IA17 share 98.2% genetic homology based on whole genomes and are located in the same monophyletic clade. Therefore, we have described in the text the two PPIV1 evaluated in this study are from the same clade of PPIV1 circulating in the US that is basically demonstrated in the phylogenetic tree in figure 1. In addition, including another figure to an already long paper would not add additional information regarding the genetic differences between PPIV1 MN16 and IA17. The author is in the process of evaluating genetic differences of PPIV1 in swine in the US, which will constitute a different manuscript when completed.

Reviewer 4 Report
The authors challenged three-week-old nursery pigs with porcine parainfluenza virus 1 (PPIV1) isolate MN16, PPIV1 isolate IA17, and a human parainfluenza virus 1 (HPIV1) strain C35 ATCC® VR-94™ to study their pathogenicity, replication kinetics, and antibody response.
Minor revision:
Page 3, line 100: the authors must inform the meaning of “DI water”.
Page 3, lines 109 - 112: the authors must clarify whether the goat anti-mouse polyclonal antibody used was biotinylated. If this is the case, the authors must inform the use of streptavidin conjugated horseradish peroxidase.
Page 3, line 127: the authors must inform the whole virus concentration used in the wv-ELISA, as well as the solution used to prepare the antigen.
Page 7, Table 3; page 10, Figure 4 legend: the authors must format Table 3 and Figure 4 legend.
Author Response
Ref.: Manuscript ID: vetsci-2092654
December 19, 2022
Dear Dr. Dashielle Li:
Thank you for considering our manuscript entitled “Characterization of two porcine parainfluenza virus 1 isolates and human parainfluenza virus 1 infection in weaned nursery pigs” for publication in Veterinary Sciences.
The reviewers indicated a series of considerations that we have addressed in the response to reviewers provided below. In addition, we have revised the manuscript extensively per the reviewer’s suggestions and to add more clarity to the results presented in the manuscript. Additionally, figure 6 was updated to provide the magnification of the images, provide the scale bars for each image, and to provide examples of lesion scores for the micrographs presented as examples of the lesions described in the tissues.
The specific changes are detailed in the response to reviewers document and in the tracked changes version of the revised manuscript. Thank you again for your time and constructive evaluation of our paper.
Sincerely,
Phillip Gauger
Iowa State University
Our responses to the reviewers are presented in italics and start with “A:”. Line numbers in reviewer comments reference the original submission; line numbers in our comments correspond to the revised manuscript with track changes.
Reviewer 4: Comments and Suggestions for Authors
The authors challenged three-week-old nursery pigs with porcine parainfluenza virus 1 (PPIV1) isolate MN16, PPIV1 isolate IA17, and a human parainfluenza virus 1 (HPIV1) strain C35 ATCC® VR-94™ to study their pathogenicity, replication kinetics, and antibody response.
Minor revision:
Page 3, line 100: the authors must inform the meaning of “DI water”.
A: The author replaced the acronym DI to deionized at lines 110.
Page 3, lines 109 - 112: the authors must clarify whether the goat anti-mouse polyclonal antibody used was biotinylated. If this is the case, the authors must inform the use of streptavidin conjugated horseradish peroxidase.
A: Two different immunocytochemistry assays were used to detect PPIV1 or HPIV1 in cell culture. The PPIV1 immunocytochemistry assay used a biotinylated goat anti-mouse IgG as a secondary antibody with a third step that used a streptavidin-HRP conjugate to help amplify the signal. The different assay was implemented for the HPIV1 immunocytochemistry assay that used a primary mouse anti-HPIV1 antibody obtained commercially, and a secondary goat anti-mouse polyclonal antibody conjugated to HRP. An additional amplification step was not needed as described for the PPIV1 immunocytochemistry assay. Additional details regarding the immunocytochemistry assays were added to lines 99-110 and 118-123.
Page 3, line 127: the authors must inform the whole virus concentration used in the wv-ELISA, as well as the solution used to prepare the antigen.
A: Because the author used concentrated virus via ultracentrifugation as antigen for coating rather than a virus lysate from a pure culture, it was not possible to estimate the protein concentration. Measuring protein concentration from the PPIV1 viral particle concentrate was attempted, but was not successful as a predictor to coat plates with antigen or to standardize viral batches for ELISA antigen. Ideally, the author would like to quantify viral particles in the preparation but there is no method to perform this with success and it is not a practical approach to designing the wv-ELISA assay.
A: However, propagating PPIV1 in cell culture was successful and the author was able to get consistent titers (similar TCID50/mL) among different batches of virus, and the author followed same procedure for concentration and dilution of the virus pellet for all ELISA plates. This translated to a similar final dilution of whole virus antigen for coating plates (1:200). In addition, the author used a checker-board approach for each batch of virus to accurately calculate the dilution necessary to coat different batches of plates (same TCID50/mL). The author published a manuscript describing the PPIV1 wv-ELISA antibody detection assay: Welch M, Krueger K, Zhang J, Piñeyro P, Magtoto R, Wang C, Giménez-Lirola L, Strait E, Mogler M, Gauger P. Detection of porcine parainfluenza virus type-1 antibody in swine serum using whole-virus ELISA, indirect fluorescence antibody and virus neutralizing assays. BMC Vet Res. 2022 Mar 21;18(1):110. doi: 10.1186/s12917-022-03196-6. PMID: 35313864; PMCID: PMC8935814.
A: The manuscript was updated with the following at lines 147-153: “The optimum dilution of concentrated PPIV1 was determined using a checkerboard titration based on known antibody positive and negative sera to maximize signal while minimizing background noise. Polystyrene 96-well ELISA plates were coated with 100 µL of the whole virus solution at an optimum dilution (1:200 in PBS) per well”
Page 7, Table 3; page 10, Figure 4 legend: the authors must format Table 3 and Figure 4 legend.
A: Per the reviewer’s suggestion, the author formatted Table 3 and Figure 4 legends and will work with the journal editor to correct any additional formatting errors that may remain in the tables or figures.

Round 2
Reviewer 3 Report
A manuscript (vetsci-2092654) entitled “Characterization of two porcine parainfluenza virus 1 isolates and human parainfluenza virus 1 infection in weaned nursery pigs” by M. Welch et al has revised almost properly by the authors.
However, this reviewer still has several minor concerns about the way of revision and would like to ask the authors the clarification once more before accepting for the publication in Veterinary Sciences.
1. Line 96; A sentence “PPIV1 primary monoclonal antibody targeting the whole virus” is logically strange, because monoclonal antibody can react only one epitope of protein and cannot react the whole virus. Please consider to rephrase like “primary monoclonal antibody mixture targeting PPIV proteins”
2. Lines 139-140; A sentence “The clarified MN16 PPIV1 supernatant was ultracentrifuged 3 times at 140,992xg for 3h, washed twice with 50 mL of PBS pH 7.4” is still not clear. Could the authors clarify why the ultracentrifugation was carried out 3 times? This reviewer does not understand the necessity of the repeat of ultracentrifugation. Please consider to rephrase like “The clarified MN16 PPIV1culture supernatant was ultracentrifuged at 140,992xg for 3h, and the pellet was rinsed twice with 50 mL of PBS pH 7.4”.
3. Figure 2; Please add the meaning of “A”, “AB” and “B” shown in Figure 2A to the figure 2 caption like “Different letters on the bars show the significancy of difference. “A” and “A” means no statistical significance (p>0.05), “A” and “AB” means the moderate statistical significance (0.01<p<0.05), and “A” and “B” means the high statistical significance (p<0.01).”
4. Figure 4; As the figure 4 caption is separated by the insertion of figure(table?) 4D, 4E and 4F, please move them to the proper position.
5. Line 433; A term “Porcine parainfluenza virus 1” is remained as the spell-out form by the reason of the authors that it is the top sentence of Discussion section. In the Instruction for Authors of Veterinary Sciences, it is described that “Acronyms/Abbreviations/Initialisms should be defined the first time they appear in each of three sections: the abstract; the main text; the first figure or table. When defined for the first time, the acronym/abbreviation/initialism should be added in parentheses after the written-out form.” As the Discussion section is contained in the main text, this reviewer thinks that it does not need the spell-out form. If the authors say about the general rule, this reviewer can find several sentences in this manuscript starting from the abbreviation form, for example PPIV1 (line 49), PPIV1 and HPIV1 virus re-isolation (line 107), PPIV1 antibody (line 134), PPIVI RNA (line 228).
Author Response
Ref.: Manuscript ID: vetsci-2092654 second revision
December 22, 2022
Dear Dr. Dashielle Li:
Thank you for considering our revised manuscript entitled “Characterization of two porcine parainfluenza virus 1 isolates and human parainfluenza virus 1 infection in weaned nursery pigs” for publication in Veterinary Sciences.
Reviewer #3 indicated additional concerns, which the authors have addressed in our response shown below. Additionally, figure 2 was updated to provide an explanation of the connecting letters found above each bar in the graph of the figure.
The specific changes are detailed in the response to reviewers document and in the tracked changes version of the revised manuscript. Thank you again for your time and constructive evaluation of our paper.
Sincerely,
Phillip Gauger
Iowa State University
Our responses to the reviewers are presented in italics and start with “A:”. Line numbers in reviewer comments reference the original submission; line numbers in our comments correspond to the revised manuscript with track changes.
Reviewer 3: Comments and Suggestions for Authors
A manuscript (vetsci-2092654) entitled “Characterization of two porcine parainfluenza virus 1 isolates and human parainfluenza virus 1 infection in weaned nursery pigs” by M. Welch et al has revised almost properly by the authors. However, this reviewer still has several minor concerns about the way of revision and would like to ask the authors the clarification once more before accepting for the publication in Veterinary Sciences.
- Line 96; A sentence “PPIV1 primary monoclonal antibody targeting the whole virus” is logically strange, because monoclonal antibody can react only one epitope of protein and cannot react the whole virus. Please consider to rephrase like “primary monoclonal antibody mixture targeting PPIV proteins”.
A: The sentence was updated per the request of the reviewer. This is updated at line 101 in the version of the manuscript provided to the author and also highlighted in yellow.
- Lines 139-140; A sentence “The clarified MN16 PPIV1 supernatant was ultracentrifuged 3 times at 140,992xg for 3h, washed twice with 50 mL of PBS pH 7.4” is still not clear. Could the authors clarify why the ultracentrifugation was carried out 3 times? This reviewer does not understand the necessity of the repeat of ultracentrifugation. Please consider to rephrase like “The clarified MN16 PPIV1culture supernatant was ultracentrifuged at 140,992xg for 3h, and the pellet was rinsed twice with 50 mL of PBS pH 7.4”.
A: The sentence was updated per the request of the reviewer. This is updated at lines 144-146 in the version of the manuscript provided to the author and also highlighted in yellow.
- Figure 2; Please add the meaning of “A”, “AB” and “B” shown in Figure 2A to the figure 2 caption like “Different letters on the bars show the significancy of difference. “A” and “A” means no statistical significance (p>0.05), “A” and “AB” means the moderate statistical significance (0.01<p<0.05), and “A” and “B” means the high statistical significance (p<0.01).”
A: The author agrees with the reviewer that a statement explaining the letters above each bar in the graph needs an explanation in figure 2, which was missing from the current version of the manuscript. This statement was added to Figure 2A at lines 317-318: “Different letters above the bar in the graph indicates a significant difference (p<0.05) between experimental groups.” However, the explanation provided by the reviewer ““A” and “AB” means the moderate statistical significance (0.01<p<0.05), and “A” and “B” means the high statistical significance (p<0.01).” is incorrect.
The “A” and “AB” does not indicate a moderate statistical difference. Because they have the same letter “A”, they are not significantly different. In addition, “A” and “B” does not indicate a high statistical significance. It only indicates a significant difference at p<0.05. For an explanation of the connecting letters reports, please see Hans-Peter Piepho (2004) An Algorithm for a Letter-Based Representation of All-Pairwise Comparisons, Journal of Computational and Graphical Statistics, 13:2, 456-466, DOI: 10.1198/1061860043515.
A: Note that in Figure 2A shown below, the Ctrl group and HPIV1 group, both with “A”, are not statistically different. In addition, HPIV1 and MN16, both with “B” above the bar graph, are also not statistically different from each other because they have the same letter “B”. However, Ctrl and IA17 are statistically different from MN16 because they do not share the same letter above the bar graph. Therefore, the author will need to keep the explanation as stated above “Different letters above the bar in the graph indicates a significant difference (p<0.05) between experimental groups.” For a similar example, please see Figure 4A in Rajão DS, Gauger PC, Anderson TK, Lewis NS, Abente EJ, Killian ML, Perez DR, Sutton TC, Zhang J, Vincent AL. Novel Reassortant Human-Like H3N2 and H3N1 Influenza A Viruses Detected in Pigs Are Virulent and Antigenically Distinct from Swine Viruses Endemic to the United States. J Virol. 2015 Nov;89(22):11213-22. doi: 10.1128/JVI.01675-15. Epub 2015 Aug 26. PMID: 26311895; PMCID: PMC4645639.
Figure 2A from the manuscript.
- Figure 4; As the figure 4 caption is separated by the insertion of figure(table?) 4D, 4E and 4F, please move them to the proper position.
A: The author apologizes for this error, but it appears to be a formatting issue that occurs when the revised manuscript is uploaded to the journal system for review. The author has attempted to correct this issue and will work with the journal editors to ensure the formatting is correct.
- Line 433; A term “Porcine parainfluenza virus 1” is remained as the spell-out form by the reason of the authors that it is the top sentence of Discussion section. In the Instruction for Authors of Veterinary Sciences, it is described that “Acronyms/Abbreviations/Initialisms should be defined the first time they appear in each of three sections: the abstract; the main text; the first figure or table. When defined for the first time, the acronym/abbreviation/initialism should be added in parentheses after the written-out form.” As the Discussion section is contained in the main text, this reviewer thinks that it does not need the spell-out form. If the authors say about the general rule, this reviewer can find several sentences in this manuscript starting from the abbreviation form, for example PPIV1 (line 49), PPIV1 and HPIV1 virus re-isolation (line 107), PPIV1 antibody (line 134), PPIVI RNA (line 228).
A: This was updated per the request of the reviewer. This is updated at line 453 in the version of the manuscript provided to the author and also highlighted in yellow.